# Segmentation of the zebrafish axial skeleton relies on notochord sheath cells and not on the segmentation clock

Laura Lleras Forero[1,2,3], Rachna Narayanan[4], Leonie FA Huitema[3‡], Maaike VanBergen[1§], Alexander Apschner[3], Josi Peterson-Maduro[3], Ive Logister[3], Guillaume Valentin[4], Luis G Morelli[5,6,7†], Andrew C Oates[4,8,9†*], Stefan Schulte-Merker[1†*]

[1]Institute for Cardiovascular Organogenesis and Regeneration, Faculty of Medicine, WWU Münster, Münster, Germany; [2]CiM Cluster of Excellence (EXC-1003-CiM), Münster, Germany; [3]Hubrecht Institute-KNAW & UMC Utrecht, Utrecht, Netherlands; [4]The Francis Crick Institute, London, United Kingdom; [5]Instituto de Investigación en Biomedicina de Buenos Aires (IBioBA), CONICET–Partner Institute of the Max Planck Society, Buenos Aires, Argentina; [6]Departamento de Física, FCEyN, UBA, Ciudad Universitaria, Buenos Aires, Argentina; [7]Department of Systemic Cell Biology, Max Planck Institute for Molecular Physiology, Dortmund, Germany; [8]Department of Cell and Developmental Biology, University College London, London, United Kingdom; [9]Institute of Bioengineering, École Polytechnique Fédérale de Lausanne (EPFL), Lausanne, Switzerland

*For correspondence:
andrew.oates@epfl.ch (ACO);
Stefan.Schulte-Merker@
ukmuenster.de (SS-M)

†These authors contributed
equally to this work

Present address: ‡Sidney
Kimmel Medical College at
Thomas Jefferson University,
Department of Dermatology and
Cutaneous Biology, The Joan
and Joel Rosenbloom Center for
Fibrotic Diseases, Philadelphia,
United States; §Department of
Laboratory Medicine, Laboratory
of Hematology, Radboud
University medical center,
Nijmegen, Netherlands

Competing interests: The
authors declare that no
competing interests exist.

Reviewing editor: Tanya T.
Whitfield, University of Sheffield,
United Kingdom

**Abstract** Segmentation of the axial skeleton in amniotes depends on the segmentation clock, which patterns the paraxial mesoderm and the sclerotome. While the segmentation clock clearly operates in teleosts, the role of the sclerotome in establishing the axial skeleton is unclear. We severely disrupt zebrafish paraxial segmentation, yet observe a largely normal segmentation process of the chordacentra. We demonstrate that axial *entpd5+* notochord sheath cells are responsible for chordacentrum mineralization, and serve as a marker for axial segmentation. While autonomous within the notochord sheath, *entpd5* expression and centrum formation show some plasticity and can respond to myotome pattern. These observations reveal for the first time the dynamics of notochord segmentation in a teleost, and are consistent with an autonomous patterning mechanism that is influenced, but not determined by adjacent paraxial mesoderm. This behavior is not consistent with a clock-type mechanism in the notochord.
DOI: https://doi.org/10.7554/eLife.33843.001

## Introduction

The segmented vertebral column is the hallmark of vertebrate species. In many vertebrates, each ossified metameric unit starts out as a chordacentrum, a small, mineralized, ring-like structure around the notochord. The chordacentrum is then expanded and later is comprised of a barrel-like centrum (or vertebral body) (*Figure 2—figure supplement 1*) and protruding neural and hemal arches that enclose and protect the spinal cord throughout the body axis as well as the inner organs. In amniotes, such as mammals and birds, all these elements as well as adjacent muscle and sub-cutis derive from transient embryonic segmented structures termed somites. The sclerotome occupies the bulk of the somite, whereas skin and muscle are derived from the smaller derma-myotome. The cells within the sclerotome responsible for producing and mineralizing the organic aspect of bone

(osteoid) are termed osteoblasts. In all vertebrates, somites are formed rhythmically and sequentially along the embryonic axis in a process governed by a multicellular, genetic oscillator termed the segmentation clock. Sequential waves of oscillating gene expression move through the precursor cells of the pre-somitic mesoderm (PSM) from the posterior to the anterior, where their arrest prefigures the timing and location of each newly forming somite boundary (*Masamizu et al., 2006*; *Aulehla et al., 2008*; *Soroldoni et al., 2014*; *Shimojo and Kageyama, 2016*). The segmentation clock contains a network of oscillating genes that shows differences between species, but members of the Hes/her family of transcriptional repressors appear to form a core negative feedback loop in all species examined (*Krol et al., 2011*). Mutations that disrupt the mouse clock (e.g. *Hes7* [*Bessho et al., 2001*]) cause defective somitogenesis and corresponding defects to the centra and arches of the spinal column. Similarly, surgical perturbation of somites in chick embryos produce defects in the centra and arches (*Stern and Keynes, 1987*; *Aoyama and Asamoto, 1988*). Engineered changes to the mouse clock that shorten the period of oscillations correspondingly produce more vertebral bodies (*Harima et al., 2013*). Thus, the current model in amniotes proposes that the segmentation clock is the major source of patterning information for the somites, which in turn provides the segmented organization of sclerotomal derivatives that form the individual centra and arches of the spinal column. Remarkably, although somitogenesis is similar in vertebrate classes, whether the cellular lineage(s) and mechanism of segmental patterning of the chordacentra are homologous remains unclear (*Fleming et al., 2015*).

It has been proposed that in the teleosts zebrafish (*Fleming et al., 2004*) and salmon (*Grotmol et al., 2003*; *Wang et al., 2013*) the notochord and not the sclerotome is the initial source of bone matrix for chordacentra formation, while in other teleost species the classical sclerotome-derived osteoblasts have been suggested as the main drivers of chordacentrum formation (*Inohaya et al., 2007*; *Renn et al., 2013*). Chordacentra are first observed as they mineralize as rings around the notochord, forming sequentially along the axis in a process that begins several days after somitogenesis and muscle segmentation is completed and lasts several weeks. This difference in developmental timing between the formation of segmented muscle and segmented skeleton in teleosts raises a registration problem between these mechanical elements, which amniotes do not need to solve. The zebrafish *fused somites/tbx6 (fss)* mutant, which has unsegmented paraxial mesoderm, has ectopically positioned neural and hemal arches, but was reported to form normal vertebral centra (*Fleming et al., 2004*; *van Eeden et al., 1996*). This suggests that a segmented sclerotome is necessary for proper arch development, but not for segmentation of the chordacentra. However, the zebrafish *hes6* mutant, which forms somites more slowly than wild type, makes correspondingly fewer centra (*Schröter and Oates, 2010*), suggesting that the segmentation clock can influence vertebral patterning. Indeed, gene expression studies suggest that the *tbx6* mutant retains a dynamic segmentation clock in the posterior of the PSM, but specifically fails to output the clock's information in the anterior PSM (*van Eeden et al., 1998*; *Holley et al., 2000*; *Nikaido et al., 2002*). The *tbx6*$^{-/-}$ and *hes6*$^{-/-}$ phenotypes might be reconciled if, even in the absence of morphological somitogenesis, an active clock in the posterior PSM of the *tbx6* mutant is sufficient to correctly pattern the chordacentra. Alternatively, the segmentation clock may not be required for segmental chordacentra formation, but may nevertheless be able to indirectly modulate the pattern.

In this paper, we investigate the developmental origin of segmental patterning of chordacentra in zebrafish, showing that the majority of chordacentra can form normally despite a disrupted segmentation clock. We image the segmentation dynamics of the notochord sheath cell layer directly and document a distinctive set of segmentation errors in mutants. To unify these observations, we describe a model for autonomous segmentation of the notochord and validate this by comparison of simulations to experimental data.

## Results

### Disruption of the segmentation clock in double and triple mutants

The zebrafish segmentation clock's core pacemaker circuit consists of *her1*, *her7* and *hes6*, which display partial and overlapping redundancy (*Schröter et al., 2012*). The analysis of mutant combinations was previously not possible because *her1* and *her7* are located ~10 Kb apart on chromosome 5. We generated a *her1;her7* double mutant by injecting a TALEN construct directed against *her1* in

the *her7* mutant (*Choorapoikayil et al., 2012*) and a novel *hes6* mutation also using a TALEN approach (*Figure 1—figure supplement 1*). The *her1* TALEN allele has two consecutive premature stop codons (TAA TAA). The predicted Her1 protein from the TALEN-induced mutation lacks the basic DNA-binding domain and the HLH dimerization domain, suggesting that the protein product has no functionality. It has been previously shown that the hu2526 allele is a *her7* null mutant resulting from a stop codon in the HLH domain (*Schröter et al., 2012*). Importantly, the phenotype of *her1;her7* double mutants is consistent with the phenotype of *her1, her7* double morphants and also of the b567 deletion allele (*Henry et al., 2002*; *Oates and Ho, 2002*).

To assess patterning in the PSM, we used *her7* (*Figure 1A–D'*), *her1* and *deltaC* expression (*Figure 1—figure supplement 2*), which show wave-like expression domains, as markers for the oscillation of the clock, and *mespb*, which shows expression in cells along the anterior border of two newly forming segments, for the clock's segmental output in the anterior PSM at the 10-somite stage (*Figure 1E–H*; *Figure 1—figure supplement 3A*). In *tbx6$^{-/-}$*, *her7* still oscillated posteriorly (*Figure 1B,B'*), but *mespb* was not expressed (*Figure 1F*), as expected (*Oates et al., 2005*; *Durbin et al., 2000*). In *her1;her7* double mutants, *her7* was expressed throughout the PSM (*Figure 1C,C'*), but lacked oscillatory waves, and *mespb* expression in the anterior PSM occurred in a diffuse domain, lacking segmented stripes (*Figure 1G*; *Figure 1—figure supplement 3B*), consistent with previous results using anti-sense knock-down reagents (*Oates and Ho, 2002*). Expression of other markers of segmental patterning in the anterior PSM (*papc*, *ripply1*, *ripply2*) also lacked segmental stripes (*Figure 1—figure supplement 3C–H*). This reveals that in *her1$^{-/-}$;her7$^{-/-}$*, the segmentation clock is severely disrupted throughout the PSM, and that a correspondingly disordered, non-segmental output is made in the anterior PSM. Triple mutants for *her1, her7* and *hes6* have the same expression patterns as *her1;her7* double *mutants* for all markers analyzed (*Figure 1—figure supplement 2*). Hence, and in order to simplify breeding, we focused further analyses on *her1$^{-/-}$; her7$^{-/-}$*. Finally, we examined *her1$^{-/-}$;her7$^{-/-}$;tbx6$^{-/-}$* and found that *her7* expression in the posterior PSM lacked oscillatory waves and neither *her7* (*Figure 1D,D'*) nor *mespb* (*Figure 1H*) was expressed in the anterior PSM. From these results we conclude that *her1$^{-/-}$;her7$^{-/-}$;tbx6$^{-/-}$* resembles the simple addition of the *tbx6* and *her1;her7* double mutant phenotypes; namely a severely disrupted segmentation clock that lacks any output in the anterior. In addition, bright field images at the 18-somite stage (*Figure 1I–L'*) were taken. Somite boundaries in wild type embryos are periodic and sharp (*Figure 1I,I'*). Somite boundaries could not be discerned in *tbx6* or *her1;her7;tbx6* triple mutants (*Figure 1J,J' and L,L'*). In *her1$^{-/-}$;her7$^{-/-}$* (*Figure 1K,K'*), partial boundaries were visible, but were lacking regular shape and periodic arrangement. This indicates that periodic morphology is disrupted in the mutant paraxial mesoderm. We next evaluated the presence of periodic patterns in muscle pioneers by analyzing *en2a* expression at the 20-somite stage (*Figure 1M–P'*). The normal periodic pattern is lost in *tbx6$^{-/-}$* (*Figure 1N,N'*), *her1$^{-/-}$;her7$^{-/-}$* (*Figure 1O,O'*) and *her1$^{-/-}$; her7$^{-/-}$;tbx6$^{-/-}$* (*Figure 1P,P'*), indicating that although these cell types are present in these mutants, there is no overt segmental pattern emerging directly from the PSM.

The characteristic chevrons of the larval myotome are visible using *xirp2a* as a boundary marker along the axis at 1.5 days post fertilization (dpf) (*Figure 2A,A'*). We observed strong disruption of periodic myotome boundaries in all mutants (*Figure 2B–D'* and *Figure 2—figure supplement 2*), with the severity in *tbx6$^{-/-}$* and *her1$^{-/-}$;her7$^{-/-}$;tbx6$^{-/-}$* (*Figure 2B,B' and D,D'*) equivalent, and stronger than that found in *her1$^{-/-}$;her7$^{-/-}$* (*Figure 2C,C'*). The short and scattered boundary fragments visible in *her1$^{-/-}$;her7$^{-/-}$* (*Figure 2C'*) correlate with earlier expression of *mespb* and partial somite boundaries in the PSM (*Figure 1G*, *Figure 1K,K'*), and may arise from a secondary morphogenetic effect of elongating muscle fibers (*van Eeden et al., 1998*), some of which are lost in the absence of *fss/tbx6* (*Windner et al., 2015*).

## Normal centra form in the absence of periodic paraxial patterning

After having established that *her1;her7* double mutants, *tbx6* single mutants and *her1;her7;tbx6* triple mutants display severe disruption of the segmentation clock and its output in the paraxial mesoderm, we examined to what extent these early paraxial defects were reflected in vertebral bodies of the adult. If the previously reported ability of the *fss/tbx6* mutant to form normal centra was due to the remaining segmentation clock activity in the posterior PSM, then we expected to see a strong disruption of vertebral bodies in both *her1$^{-/-}$;her7$^{-/-}$* and *her1$^{-/-}$;her7$^{-/-}$;tbx6$^{-/-}$* adults. Mineralized bone was visualized using Alizarin Red (AR) staining of adult skeletons. We observed

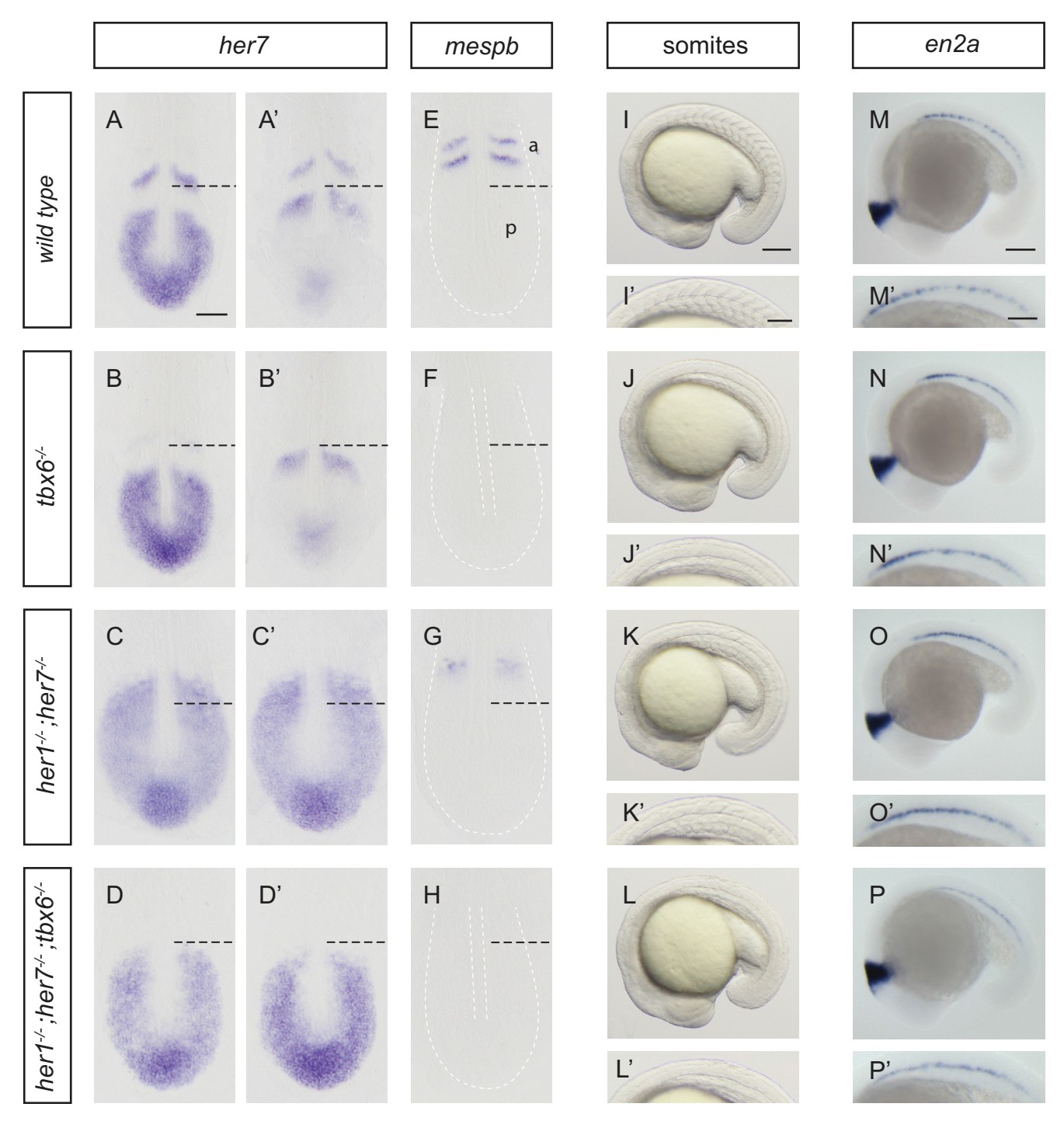

**Figure 1.** Disruption of the segmentation clock in *tbx6*, *her1;her7* and *her1;her7;tbx6* mutants. (A–D') In situ hybridization for segmentation clock marker *her7*. (B and B') *her7* oscillates in the posterior PSM of *tbx6⁻/⁻*, but does not oscillate in *her1⁻/⁻;her7⁻/⁻* (C and C') or *her1⁻/⁻;her7⁻/⁻;tbx6⁻/⁻* (D and D'). (E–H) In situ hybridization for segmental output marker *mespb*. *mespb* is not expressed in *tbx6⁻/⁻* (F) or *her1⁻/⁻;her7⁻/⁻;tbx6⁻/⁻* (H), but is weakly expressed in *her1⁻/⁻;her7⁻/⁻*, albeit not in segmental stripes (G). (I–L') Somite boundaries in the paraxial mesoderm. In *tbx6⁻/⁻* (J), *her1⁻/⁻;her7⁻/⁻* (K) and *her1⁻/⁻;her7⁻/⁻;tbx6⁻/⁻* (L) mutants, boundaries lose periodic order. (M–P') Spatial distribution of muscle pioneers marked by in situ hybridization with *en2a*. In *tbx6⁻/⁻* (N), *her1⁻/⁻;her7⁻/⁻* (O) and *her1⁻/⁻;her7⁻/⁻;tbx6⁻/⁻* (P) muscle pioneers lose segmental pattern. A-H' are dorsal views of 13.5 hpf (10 somites) embryos, I-P' are lateral views of 18–19.5 hpf (18–20 somites) embryos. a – anterior, p – posterior. Scale bar in A is 100 µm and applies to A-G. Scale bar in I is 150 µm, applies to I-L and in I' is 100 µm, applies to I'-L'. Scale bars in M and M' are 150 µm and 100 µm respectively, and apply to M-P and M'-P' respectively.

*Figure 1 continued on next page*

*Figure 1 continued*

DOI: https://doi.org/10.7554/eLife.33843.002

The following figure supplements are available for figure 1:

**Figure supplement 1.** Generating novel *her1;her7* and *hes6* mutants by TALEN.

DOI: https://doi.org/10.7554/eLife.33843.003

**Figure supplement 2.** Disruption of the segmentation clock in *tbx6*, *her1;her7*, *her1;her7;tbx6* and *her1;her7;hes6* mutants.

DOI: https://doi.org/10.7554/eLife.33843.004

**Figure supplement 3.** Disruption of segmental output in the anterior PSM of *her1;her7* mutants.

DOI: https://doi.org/10.7554/eLife.33843.005

duplications, fusions and abnormalities of the neural and hemal arches throughout the axis in every $tbx6^{-/-}$, $her1^{-/-};her7^{-/-}$ and $her1^{-/-};her7^{-/-};tbx6^{-/-}$ adult, consistent with a loss of pattern in the sclerotome. However, the majority of centra in $her1^{-/-};her7^{-/-}$ and $her1^{-/-};her7^{-/-};tbx6^{-/-}$ adults were remarkably well-formed, defined as being cleanly separated from neighboring centra and similar to wildtype in their basic hourglass shape (*Figure 2F–H*). Furthermore, we confirmed that all centra were well-formed in all *her1*, *her7* and *hes6* hetero- and homozygotes (*Hanisch et al., 2013*), in all heterozygous double and triple crosses, and in *hes6;her7* double mutants. In addition, centra were also well-formed in mutants where the segmentation clock's oscillating cells slowly desynchronize due to a loss in Delta-Notch signaling, such as in *beamter/deltaC* and *after eight/deltaD* (*Durbin et al., 2000*) (*Figure 2—figure supplement 3* and *Figure 2—figure supplement 4*). Thus, the segmentation clock activity remaining in the posterior PSM of *tbx6* mutant embryos is not the cause of the well-formed vertebral centra observed in $tbx6^{-/-}$ adults.

However, in some mutant combinations we did observe localized defects, such as a bend in the axis or occasional malformations and fusion of neighboring centra, scattered along every *her1;her7* and *her1;hes6* double mutant, and *her1;her7;hes6* and *her1;her7;tbx6* triple mutant skeletons, as well as in 80% of $tbx6^{-/-}$ skeletons (*Figure 2F–H* and *Figure 2—figure supplement 4R,U*). This shows that previous reports of normal segmentation of the centra in $tbx6^{-/-}$ were incomplete (*Fleming et al., 2004*; *van Eeden et al., 1996*), and indicates that in the absence of periodic order in the early paraxial mesoderm, formation of the centra is error-prone. Given the proximity of the developing chordacentra to the notochord, and the suggestion that the notochord serves as a linear template for the vertebral column (*Gray et al., 2014*), our findings argue instead that formation of periodic chordacentra may arise from a separate segmentation mechanism intrinsic to the notochord.

## Segmental *entpd5* expression in notochord sheath cells is the key step for chordacentrum mineralization

Secreted Entpd5 has previously been shown to be required for bone formation in zebrafish, and to be co-expressed with *osterix (osx)* in craniofacial osteoblasts (*Huitema et al., 2012*). When examining *entpd5* promoter activity outside the craniofacial area, we observed a striking segmented pattern in the sheath cells along the notochord (*Figure 3A*), well before the onset of chordacentrum mineralization is first observed in the anterior notochord at 6 dpf (*Morin-Kensicki et al., 2002*). Photoconversion at 3 dpf (*Figure 3B*) of Kaede expressed from an *entpd5:kaede* transgene showed that these rings arise from an earlier ubiquitous expression domain by de novo synthesis (*Figure 3C*), making *entpd5* the earliest axial segmented marker known for zebrafish. We found *entpd5* to be segmentally expressed first in the dorsal region of the sheath cell layer at the anterior, and new, periodically forming rings of *entpd5* expression formed sequentially from anterior to posterior along the axis (*Figure 3C and D*). *entpd5+* sheath cells precisely predicted the position of the mineralized chordacentra and were located centrally to the mineralized matrix as revealed by co-staining with AR (*Figure 3E and F*). In classical osteoblasts (cells able to produce bone matrix), as are found in the cleithrum or parasphenoid bones of the head, *entpd5* is co-expressed with the osteoblast regulator *osterix/Sp7* (*Figure 3—figure supplement 1A*) whereas within the developing vertebral column, *osterix* expression is first observed after 17 dpf when neural and hemal arches begin to form (*Spoorendonk et al., 2008*). *entpd5* mutants form normally segmented osteoid, but do not

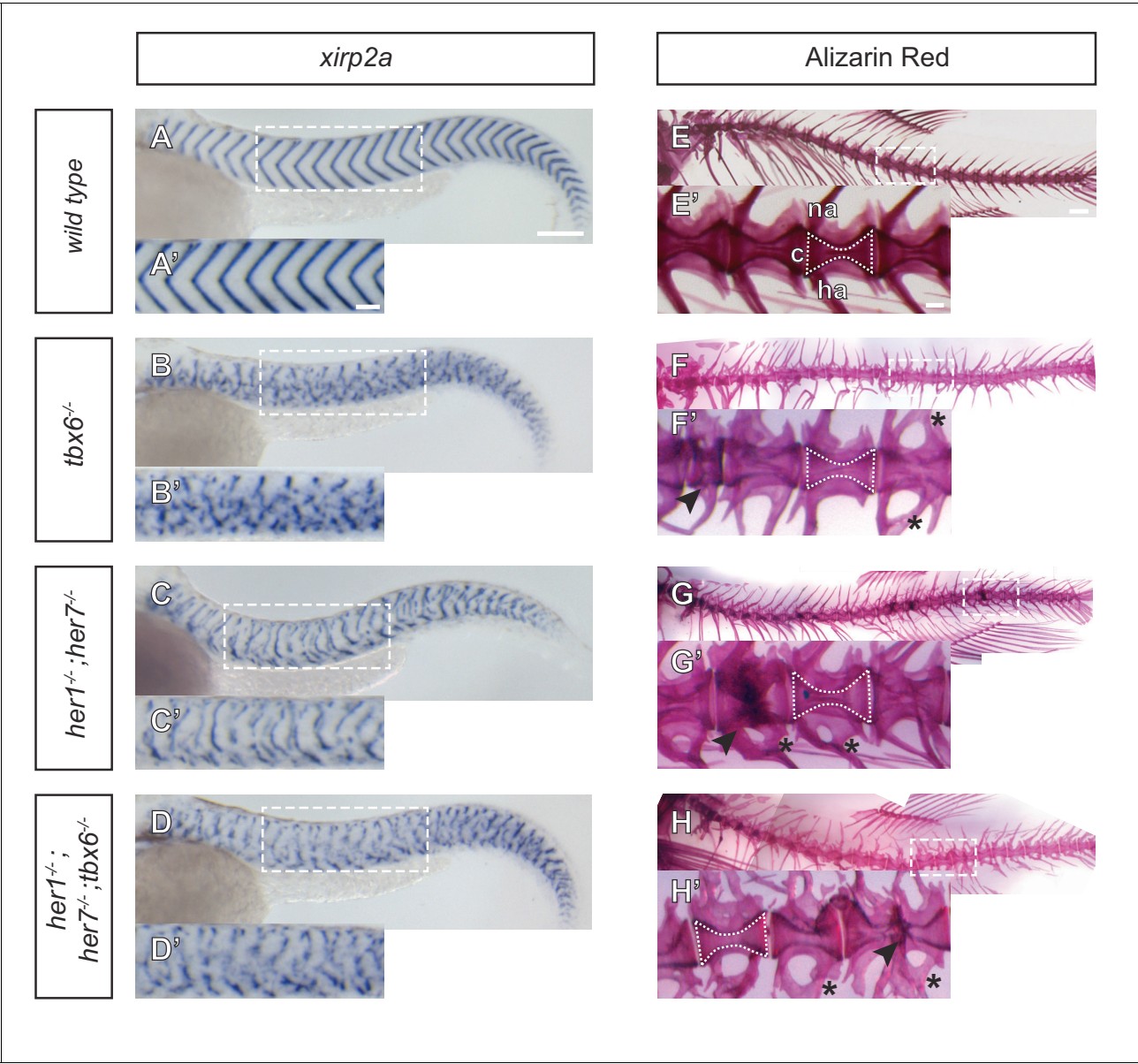

**Figure 2.** Myotome boundaries are disrupted in segmentation clock mutants, but chordacentra are still patterned. (**A** to **D'**) In situ hybridization for myotome boundary marker *xirp2a*. Myotome boundaries are disrupted to differing degrees of severity depending on the genotype. (**E–H'**) Alizarin Red bone preparations. Centra are well-formed in *tbx6*$^{-/-}$ (n = 10) (**F**), while neural and hemal arches are often fused (**F'**). Centra are also well-formed in *her1*$^{-/-}$;*her7*$^{-/-}$ (n = 14) (**G,G'**) and *her1*$^{-/-}$;*her7*$^{-/-}$;*tbx6*$^{-/-}$ (n = 15) (**H,H'**). Occasional defects occur, seen as smaller vertebrae (arrowhead in F'), or as fusions of two vertebrae (arrowheads in G' and H'). Larvae in A-D are 40 hpf. Adult fish in E-H are between two and six months. All animals in lateral view with anterior to the left. na - neural arch, hr - hemal arch, c - centrum. Scale bars in A and A' are 150 µm and 100 µm respectively and apply to A-D and A'-D' respectively. Scale bar in E is 1 mm and applies to E-H, scale bar in E' is 200 µm and applies to E'-H'. Asterisks highlight fused neural and hemal arches.

DOI: https://doi.org/10.7554/eLife.33843.006

The following figure supplements are available for figure 2:

**Figure supplement 1.** Schematic depiction of a vertebral body.

DOI: https://doi.org/10.7554/eLife.33843.007

**Figure supplement 2.** Severity of myotome boundary disruptions in *tbx6*, *her1;her7* and *her1;her7;tbx6* mutants differ according to genotype.

DOI: https://doi.org/10.7554/eLife.33843.008

**Figure supplement 3.** Centra are well-formed in *deltaD*, *deltaC*, *her1*, *her7* and *hes6* segmentation clock single gene mutants.

DOI: https://doi.org/10.7554/eLife.33843.009

**Figure supplement 4.** Segmentation clock gene double and triple heterozygous mutants have well-formed centra.

*Figure 2 continued on next page*

*Figure 2 continued*

DOI: https://doi.org/10.7554/eLife.33843.010

mineralize it (*Huitema et al., 2012*). In contrast, *osterix* mutants have normal *entpd5* expression and ossification in the notochord, but reduced ossification in the head (*Figure 3—figure supplement 1B,C*).

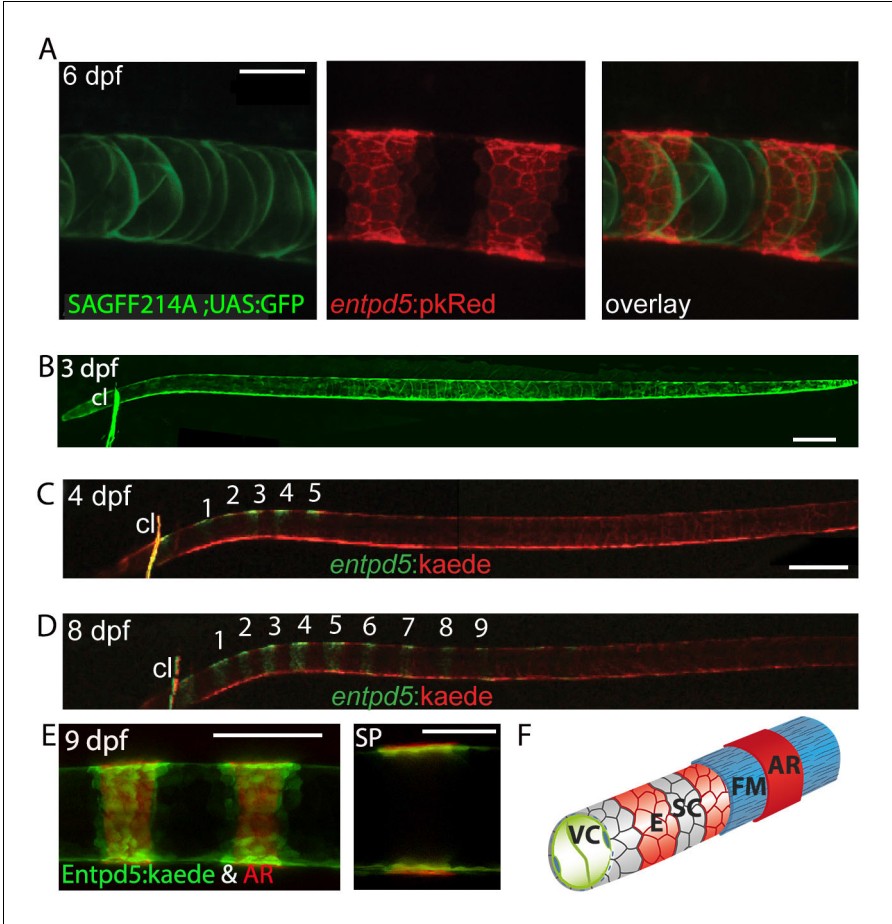

**Figure 3.** Segmental e*ntpd5* expression in notochord sheath cells marks the sites of chordacentrum mineralization. (**A-D**) Confocal images of live transgenic *entpd5* reporter larvae in lateral view with anterior to left. (**A**) At 6 dpf, *entpd5* is expressed only in notochord sheath cells and not in vacuolated notochord cells, labelled by SAGFF214A;UAS:GFP. (**B**) At 3 dpf *entpd5* is expressed in the whole notochord and does not display a segmented pattern. (**C,D**) Transgenic *entpd5:Kaede* embryos were photoconverted at 3 dpf and imaged at 4 dpf (**C**) and 8 dpf (**D**), respectively. New axial expression domains (green) are restricted to a segmental pattern within the axis and the cleithrum (cl). (**E**) Live confocal imaging of *entpd5*:Kaede expression in larvae also stained with Alizarin Red (AR) in lateral view (left) and sagittal view (SP). *entpd5+* expression domains overlap with areas of mineralization (left), and notochord sheath cells (green) localize proximal to the site of mineralization of the future chordacentra. (**F**) Schematic illustration depicting the innermost vacuolated cells (VC) and the alternating pattern of e*ntpd5+* (red, E) and *entpd5-* (grey) notochord sheath cells (SC). The sheath cells are surrounded by a fibrous matrix (FM), which in turn becomes mineralized in *entpd5+* areas. cl, cleithrum. Scale bar for A and E is 40 μm, scale bar for B and C is 150 μm.

DOI: https://doi.org/10.7554/eLife.33843.011

The following figure supplement is available for figure 3:

**Figure supplement 1.** *osterix* is required for the formation of cranial bone structures, but not for the axial skeleton.

DOI: https://doi.org/10.7554/eLife.33843.012

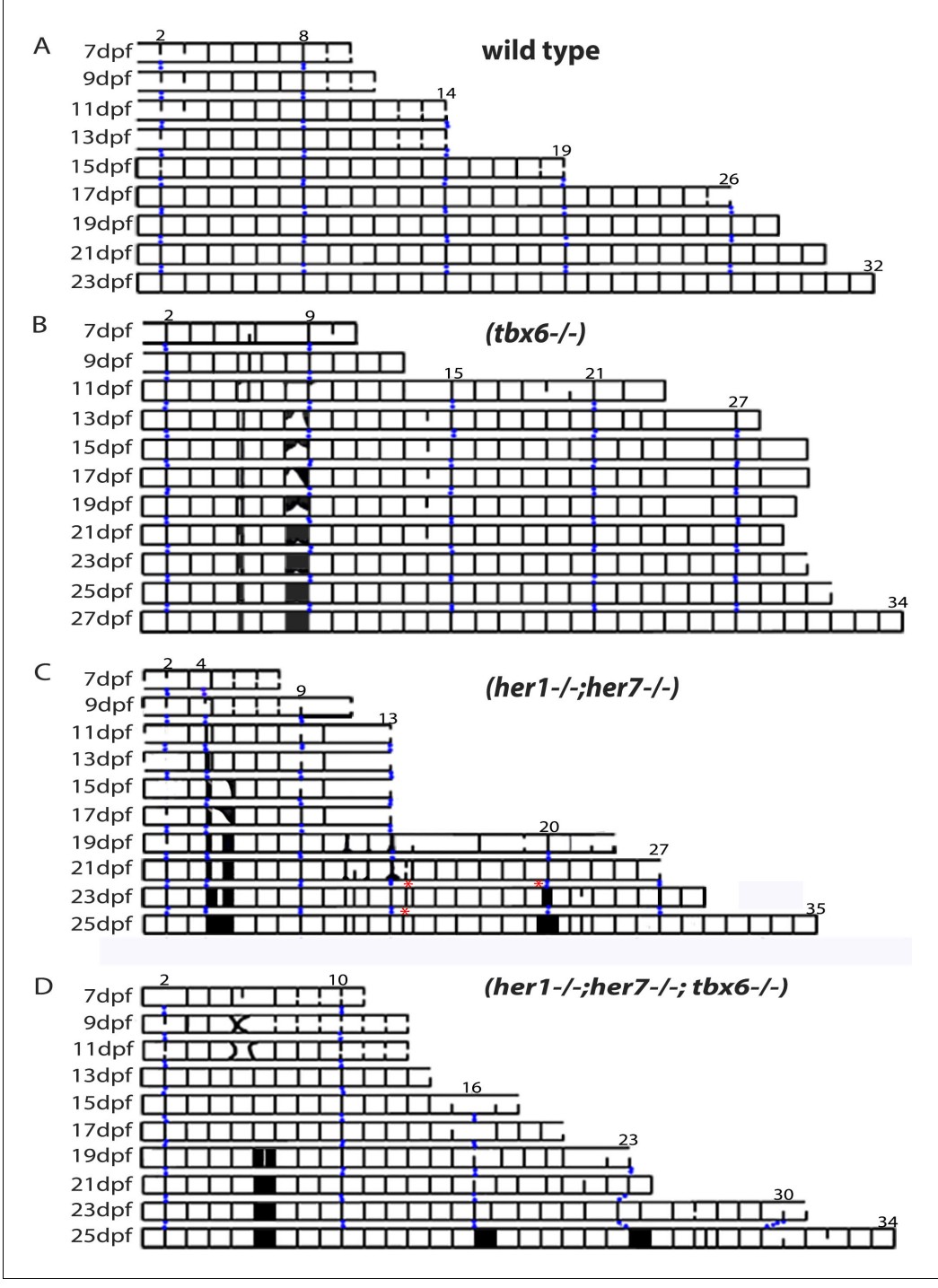

**Figure 4.** Mutants with disturbed segmentation clock form a metameric order of chordacentra with scattered defects. (**A–D**) Kymogram representation of virtual time lapse observations of representative *entpd5*:kaede-expressing larvae of each genotype. (**A**) In wild type (n = 16), *entpd5+* segments are added in an orderly manner from anterior to posterior (black lines). (**B**) *tbx6*$^{-/-}$ (n = 4), (**C**) *her1*$^{-/-}$;*her7*$^{-/-}$ (n = 4), and (**D**) *her1*$^{-/-}$;*her7*$^{-/-}$;*tbx6*$^{-/-}$ mutants (n = 4) also form *entpd5+* rings in an anterior to posterior manner, but in an error-prone fashion. Defects such as gaps in the segmental pattern (increased intervertebral spaces), or small vertebrae followed by fusions (black boxes) can be seen scattered along the axis. Red asterisks (*) in *her1*$^{-/-}$; *her7*$^{-/-}$ kymogram (**C**) represent sites of transient bending of the axis. Blue dots follow the development of the numbered chordacentra over time.

DOI: https://doi.org/10.7554/eLife.33843.013

*Figure 4 continued on next page*

*Figure 4 continued*

The following figure supplements are available for figure 4:

**Figure supplement 1.** Wild type *entpd5*:kaede larvae develop axial segmentation in an orderly manner from anterior to posterior.

DOI: https://doi.org/10.7554/eLife.33843.014

**Figure supplement 2.** *fss (tbx6$^{-/-}$);entpd5:kaede* larvae develop axial segmentation with occasional gaps that are later filled by a smaller *entpd5*+ ring domain.

DOI: https://doi.org/10.7554/eLife.33843.015

**Figure supplement 3.** *her1$^{-/-}$;her7$^{-/-}$;entpd5:kaede* larvae develop axial segmentation in a disorganized manner, occasionally missing one or two segments or inserting additional segments.

DOI: https://doi.org/10.7554/eLife.33843.016

**Figure supplement 4.** *her1$^{-/-}$;her7$^{-/-}$;tbx6$^{-/-}$;entpd5:kaede* larvae showed disorganized axial segmentation.

DOI: https://doi.org/10.7554/eLife.33843.017

**Figure supplement 5.** *hes6$^{-/-}$;entpd5:kaede* larvae segment their axis in an orderly manner.

DOI: https://doi.org/10.7554/eLife.33843.018

## The patterning mechanism in the notochord sheath is influenced, but not determined, by paraxial segmentation

We reasoned that a better understanding of the occasional defects observed in the *tbx6$^{-/-}$,her1$^{-/-}$; her7$^{-/-}$*, and *her1$^{-/-}$;her7$^{-/-}$;tbx6$^{-/-}$* chordacentra might provide insight into the mechanism of chordacentra segmentation. To describe the dynamics of these defects we first imaged *entpd5* expression in the sheath cells along the axis at intervals of 2 days in each mutant and recorded the distribution in a kymogram (*Figure 4A–D*). In wildtype larvae (*Figure 4A*; *Figure 4—figure supplement 1*), *entpd5*+ segments are established in an anterior to posterior progression at a rate of ~1.5 per day, with the exception of the first two segments (part of the Weberian apparatus) which appear dorsally at first and are completed 4–6 days later. As each chordacentrum matured, *entpd5* expression was down-regulated in the center and was retained at the distal edges of each element, as illustrated in *Figure 2—figure supplement 1* and *Figure 4—figure supplement 1*. This process began around 15 dpf in the anterior chordacentra, and sequentially progressed posterior-wards along the axis.

We next examined the dynamics of *entpd5* expression rings in different mutant backgrounds. In the case of *tbx6* (n = 4), *her1;her7* (n = 4), *her1;her7;tbx6* (n = 4) and *hes6* (n = 6) mutants (*Figure 4B–D*, *Figure 5A and B*, *Figure 4—figure supplement 2*, *Figure 4—figure supplement 3*, *Figure 4—figure supplement 4* and *Figure 4—figure supplement 5*), the chordacentra formed in an anterior to posterior direction as in . However, we observed two types of defects: in the first, a ring was initially not formed, leaving a transient gap (*Figure 5A* and longer spaces in the kymograms), which was then modified by the subsequent intercalation of a new *entpd5* expression ring two or more days later (segment 1' in *Figure 5A*, 19 dpf). The intercalated ring was thinner than neighboring elements, likely reflecting an earlier stage in ring development, and remained smaller and distinct from neighboring segments. In the second type of defect, a ring was added (on schedule or out of schedule) in the middle of a normal intervertebral distance, creating a distance shorter than expected between two rings. In this case, the notochord sheath cells of the small segment fused with one or two of its neighboring segments (*Figure 5B*, black boxes in kymograms). In the cases where two sequential defects are inserted by any of the two processes described above, we observed a transient bending of the axis, which was later modified by fusing segments or leaving two or three contiguous small vertebrae (*Figure 4C*, *her1$^{-/-}$;her7$^{-/-}$* kymogram and *Figure 4—figure supplement 3*). The phenotypic defects we see are different to the scoliosis phenotypes reported in mutant *leviathan/col8a1a* animals with defects in collagen deposition (*Gray et al., 2014*) or in mutant *stocksteif/cyp26b1* animals with increased retinoic acid levels (*Spoorendonk et al., 2008*). In these cases, the first pattern of mineralization appears correctly segmented, and defects arise because of axial bending of the notochord and bone overgrowth, respectively. In contrast, we have directly observed the initial emergence of defective *entpd5*+ rings, suggesting that the periodic patterning of the notochord has been affected.

Segment defects in the paraxial mesoderm occur sequentially along the axis, consistent with a disrupted clock-type mechanism. In contrast, the intercalation defects observed in the mutant axes

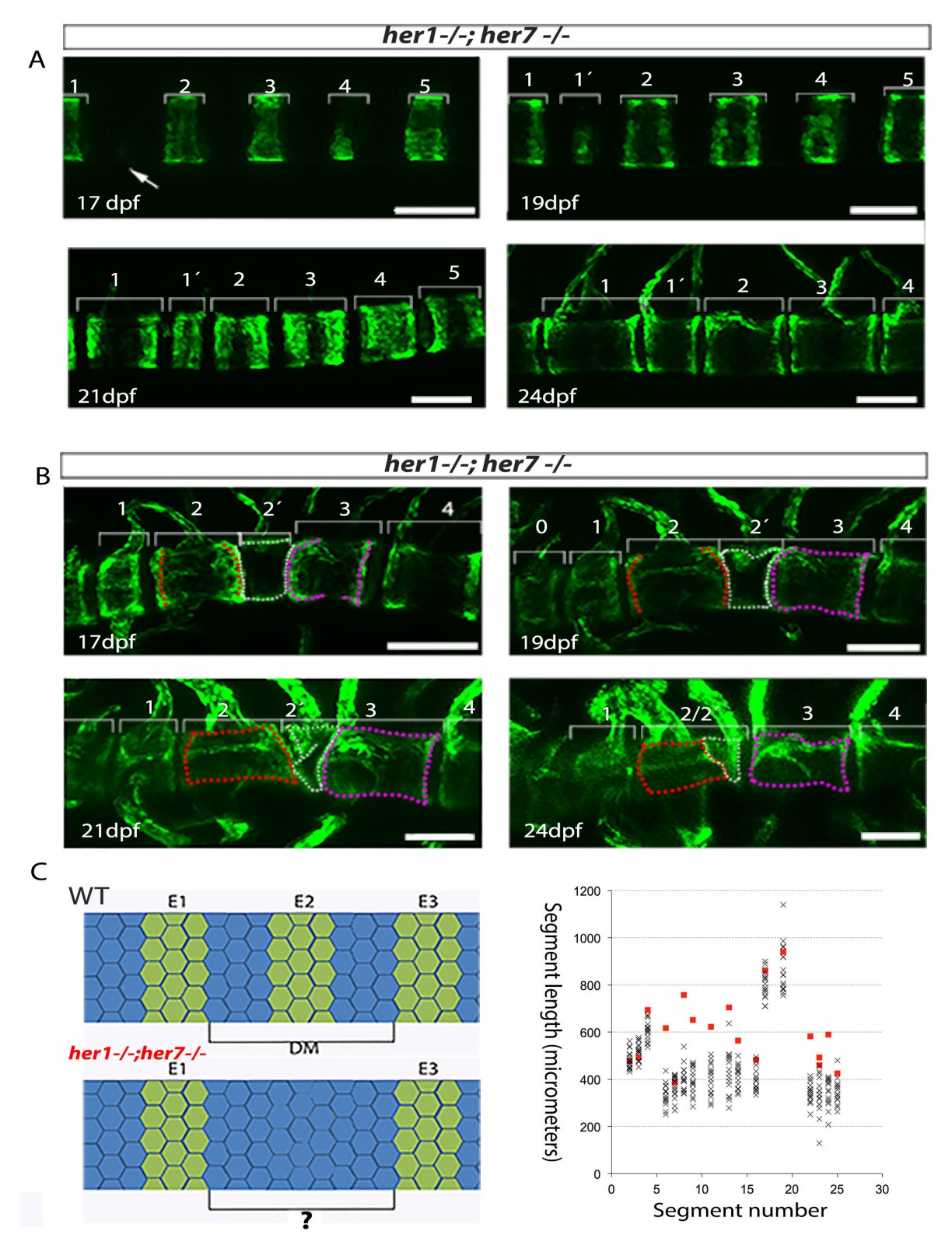

**Figure 5.** Inaccurate spacing of *entpd5+* segments results in erroneous chordacentrum formation. (**A, B**) Time series images of *entpd5+* segments around the notochord in *her1;her7* mutants, in lateral view with anterior to the left. (**A**) An atypically wide space between *entpd5+* segments (arrow) results in the subsequent intercalation of an additional, smaller *entpd5+* segment (1′). (**B**) An additional smaller segment (2′) fuses to adjacent vertebra. (**C**) The length between existing *entpd5+* segments was measured in *her1;her7* mutants (n = 4) in positions where an *entpd5* ring would be intercalated

*Figure 5 continued*

(red dots) and compared to the equivalent axial position in wild type (WT, n = 16) (black crosses). The distance preceding an intercalation in *her1;her7*
mutants was either similar or larger than wild type. DM, distance measured; E *entpd5*+ segment. All scale bars are 100 μm.

DOI: https://doi.org/10.7554/eLife.33843.019

form out of schedule with the sequence of notochord segmentation. We asked if the intercalation of an *entpd5* ring was associated with an error in the initial local spacing of the rings along the notochord. We measured the distance between the *entpd5*+ rings bordering the location where the intercalated segment was added at the time point immediately before its appearance in *her1;her7* mutants (n = 4 animals; 18 intercalations, red squares; *Figure 5C*) and compared this to equivalent distances in control embryos (n = 16 animals, 288 segments, black crosses). The distance preceding an intercalation was either similar to or larger than expected from the controls, but never smaller (n = 18, 1.4 ± 0.3 fold increase, mean ± SD). Thus, the intercalatory rings formed at a range of distances from their earlier neighbors, suggesting that the defects arise as a response to an error in positioning the earlier rings and are not a simple delay in *entpd5* expression. These distinctive intercalations are difficult to reconcile with a clock-type mechanism for segmenting the notochord.

The chordacentra defects described above may reflect some influence of the disrupted paraxial mesoderm pattern on the notochord. Previously, it was shown that *hes6* mutants show weak myotome boundary defects in the posterior tail at low penetrance at 1.5 dpf (*Schröter and Oates, 2010*), and the *hes6* mutant generated in this paper also show these defects (9/67 (13.4%)). Nevertheless, the existence and distribution of skeletal defects had not been examined. In *hes6*$^{-/-}$ skeletons we observed from one to four centra defects, restricted to the caudal vertebrae, in AR stains of adult bone (4/15, 27%) (*Figure 6A and B*) and in *entpd5* expression at 28 dpf (8/41, 20%) (*Figure 6C–E*). Additional analysis showed that 2/5 (40%) *hes6* mutants with a chordacentra defect revealed by *entpd5* expression also presented a myotome defect in the same region. The remaining 60% (3/5) had no myotome defect, but still showed a chordacentra defect. This partial overlap of myotome and chordacentra defects in *hes6* mutants, as well as the coordinated reduction in total myotome and vertebral number observed previously (*Schröter and Oates, 2010*) and in our work suggests that the pattern of the paraxial mesoderm can influence the segmentation of the notochord.

## A reaction-diffusion model of axial patterning in the zebrafish

These findings can be synthesized in a model of axial segmentation in which the notochord possesses an intrinsic segmentation mechanism, likely within the sheath cells, that does not depend on the paraxial segmentation clock to produce periodic *entpd5* rings and subsequent mineralization. This mechanism is proposed to act directly upstream of *entpd5* expression, but is nevertheless sensitive to information from the paraxial mesoderm, likely from the myotome structure, which can bias the position of a ring to enable coupling of the early-developing myotome with the later-forming skeleton in wildtype.

To assess the plausibility of this hypothesis and to investigate what kind of mechanism has these properties, we developed a theoretical description that formalizes these ideas and incorporates key experimental findings. We describe the intrinsic patterning mechanism operating in the notochord sheath cells as a reaction diffusion system with two components, an activator and an inhibitor (*Murray, 1993*) (see the Theory section in Materials and methods and *Figure 7—figure supplement 1*). This theory is capable of producing an autonomous pattern, sequentially adding segments from anterior to posterior (*Figure 7A*, *Figure 7—figure supplement 2* and *Video 1*). The cues provided by the paraxial mesoderm pattern are introduced as a distribution of sinks for the inhibitor that are of the same order as the mechanism's intrinsic wavelength (*Figure 7*). The choice of sinks instead of sources, together with vanishing initial conditions across the notochord except for a perturbation localized at the anterior, are required to preserve the sequential character of notochord segmentation. Although other more complex descriptions that improve robustness to noise are possible (Materials and methods), this simple theory successfully accounts for the sequential formation of regular segments in the presence of sinks as observed in wild type (*Figure 7B*, *Figure 7—figure supplement 3* and *Video 2*). The intrinsic patterning mechanism allows for a range of pattern wavelengths,

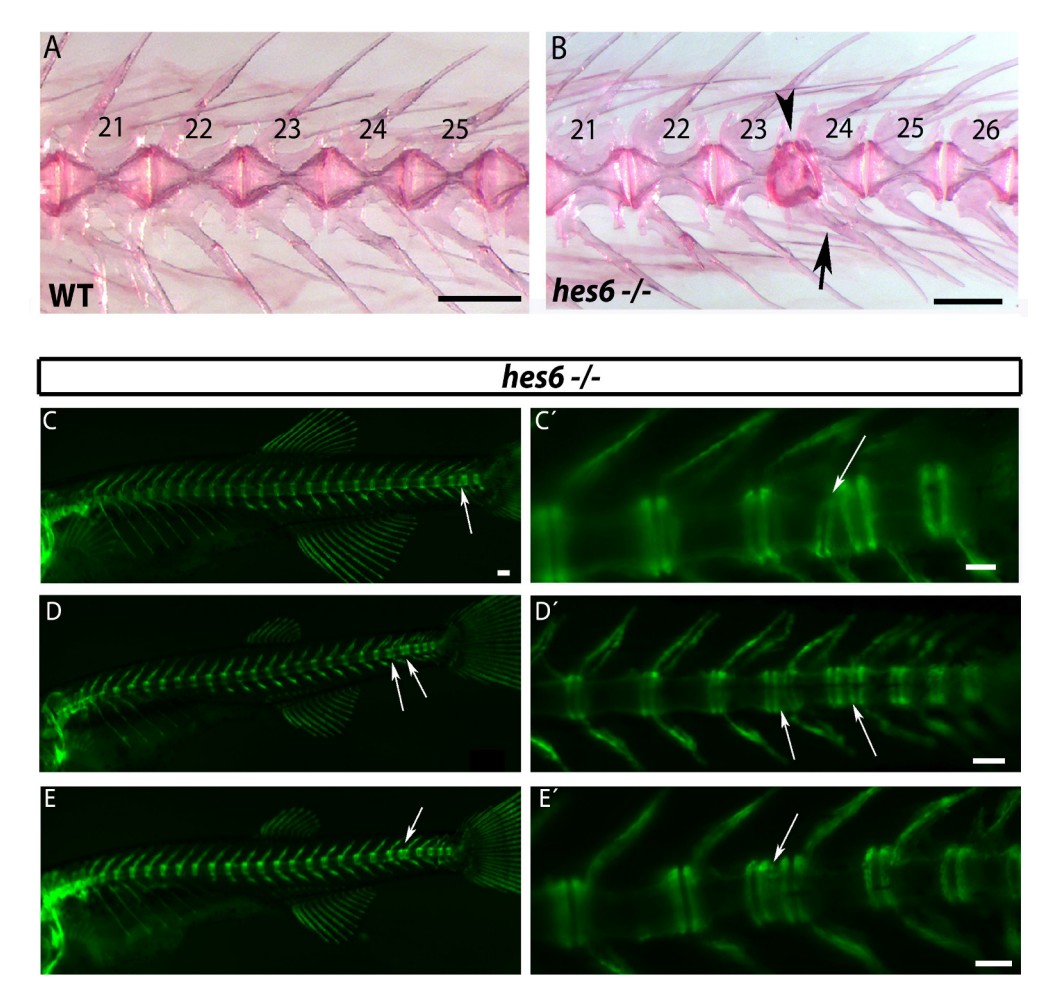

**Figure 6.** *hes6* mutant embryos can form defective caudal vertebrae. (A,B) Alizarin Red bone preparations of wild type and *hes6⁻/⁻* adults. (B) 27% of *hes6⁻/⁻* adult bone stains presented with defects in caudal chordacentra (n=4/15) wildtype. Arrow points at fused hemal arches, arrow head at chordacentra segment defect. (C to E′) *entpd5*: YFP expression in *hes6* mutants at 28 dpf . 20% of *hes6* mutants have one or more defective small vertebrae (arrows) exclusively in the caudal axis (n=8/41). Scale for A and B is 2.5 mm, C is 300 μm and C′ to E′ is 200 μm.
DOI: https://doi.org/10.7554/eLife.33843.020

providing the plasticity for the sinks to pin segments to specific locations, altering the length of each segment. We conjecture that the mutants do not affect the intrinsic notochord patterning mechanism, but change the features of the sink distribution. The potential of sinks for biasing the pattern can interfere with ring formation, revealing a process that can intercalate a ring into a mispatterned gap in the sequence. Both reducing the sink strength and increasing the noise in the positioning of sinks can induce defects in the intrinsic patterning mechanism (*Figure 7—figure supplement 4*). Thus, the spatially disordered and variable sized myotomes observed in the mutants (*Figure 2A–E*) are described in the theory as large fluctuations in the positions and amplitudes of inhibitor sinks (*Figure 7C and D*). The *her1⁻/⁻;her7⁻/⁻* situation is described by strong sinks with large position errors (*Figures 2C* and *7B*, *Figure 7—figure supplement 5* and *Video 3*) while *tbx6⁻/⁻* and *her1⁻/⁻;her7⁻/⁻;tbx6⁻/⁻* are described by sinks with reduced amplitude and shorter wavelength, accounting for the smaller and scattered myotome fragments (*Figures 2B,D and* and *7D*, *Figure 7—figure supplement 6* and *Video 4*). In the framework of the theory, *tbx6⁻/⁻* and *her1⁻/⁻;her7⁻/⁻;tbx6⁻/⁻* are described by the same set of parameters. The mechanism has some flexibility and is compatible with a range of wavelengths. For example, the larger myotomes

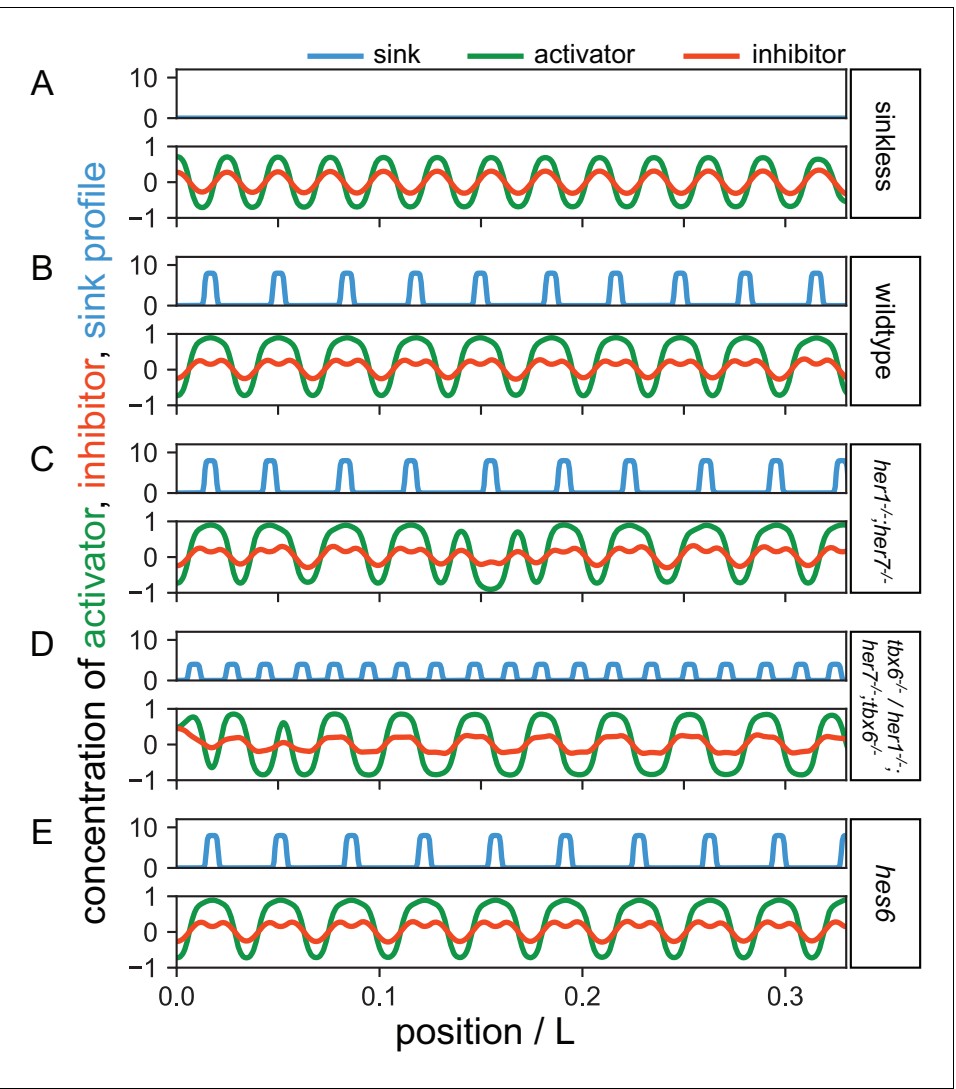

**Figure 7.** A reaction diffusion theory accounts for key experimental findings. A sink profile (blue) describes cues from myotomes that bias the position of segments. The Entpd5 pattern is given by the concentration of an activator (green) that is regulated by an inhibitor (red). (A) The system is capable of autonomous pattern formation in the absence of sinks. (B) Wild type condition is described by regularly placed strong sinks, according to the output of a functioning segmentation clock. (C) In $her1^{-/-};her7^{-/-}$ strong sinks are misplaced due to a malfunctioning segmentation clock, causing segments to be also misplaced and giving rise to defects. (D) $tbx6$ and $her1;her7;tbx6$ mutants are characterized by weaker segmentation clock output and fragmented and scattered myotome boundaries, here described by weaker sinks with a shorter wavelength. (E) The $hes6$ mutant is here characterized by a sink profile wavelength that is 6% larger than wild type. Parameters: $a = 10^{-3}$, $b = 10^{-2}$, $\tau = 0.1$, $d = 0.5$. Sink profile parameters: (A) $S_0 = 0$, (B) $S_0 = 8$, $\lambda = 0.57$, $\sigma = 0.05$, (C) $S_0 = 8$, $\lambda = 0.57$, $\sigma = 0.18$, (D) $S_0 = 4$, $\lambda = 0.30$, $\sigma = 0.15$, and (E) $S_0 = 8$, $\lambda = 0.60$, $\sigma = 0.05$. See also *Videos 1–5* and *Figure 7—figure supplement 1–9* for animations and snapshots for all conditions. Source data files for this figure and the supplemental figures have been supplied.

DOI: https://doi.org/10.7554/eLife.33843.026

The following figure supplements are available for figure 7:

**Figure supplement 1.** Theory schematics.
DOI: https://doi.org/10.7554/eLife.33843.027

**Figure supplement 2.** Sequence of snapshots from *Video 1* showing simulation of the autonomous sinkless condition, in which patterning occurs sequentially from anterior to posterior.
DOI: https://doi.org/10.7554/eLife.33843.028

**Figure supplement 3.** Sequence of snapshots from *Video 2* showing simulation of the wild type condition.
*Figure 7 continued on next page*

*Figure 7 continued*

DOI: https://doi.org/10.7554/eLife.33843.029

**Figure supplement 4.** Theoretical effects of sink strength and sink wavelength noise on notochord patterning mechanism.

DOI: https://doi.org/10.7554/eLife.33843.030

**Figure supplement 5.** Sequence of snapshots from *Video 3* showing simulation of the *her1;her7* mutant condition.

DOI: https://doi.org/10.7554/eLife.33843.031

**Figure supplement 6.** Sequence of snapshots from *Video 4* showing simulation of the *tbx6* mutant condition.

DOI: https://doi.org/10.7554/eLife.33843.032

**Figure supplement 7.** Sequence of snapshots from *Video 5* showing simulation of the *hes6* mutant condition.

DOI: https://doi.org/10.7554/eLife.33843.033

**Figure supplement 8.** Chordacentra always align with the myotome boundaries in wild type larvae, but not in mutants.

DOI: https://doi.org/10.7554/eLife.33843.034

**Figure supplement 9.** Quantification of mutant phenotype observables and comparison to theoretical description.

DOI: https://doi.org/10.7554/eLife.33843.035

produced by a *hes6* mutant provides larger wavelength cues to the notochord resulting in larger and fewer chordacentra (*Figure 7E*, *Figure 7—figure supplement 7* and *Video 5*).

In the simulations of the wild type, there is always a correspondence between the position of the sink and the position of the peak of activator (*Figure 7B and E*); this situation is also found between the positions of the myotome boundary and the *entpd5* expression ring in experimental wild type animals (*Figure 7—figure supplement 8A–F*). In the simulations of *tbx6*, *her1;her7* and *her1;her7; tbx6 mutants* this strict correspondence between sink and activator is lost (*Figure 7C and D*); activator peaks occur both together with sinks and in between them. In the case of *hes6*, the strict correspondence between sink and activator may be lost in the last tail segments. To test this prediction of the model, we examined the distribution of myotome boundaries and *entpd5* expression rings in *tbx6* and *her1;her7* double mutants. We observed that the disorganized myotome boundary fragments in the mutants had lost strict correspondence with the *entpd5* rings (*Figure 7—figure supplement 8G–R*). This lack of spatial correspondence between paraxial structure and axial expression is in agreement with the model, and it supports the hypothesis that the sink is associated with some feature of the myotome boundary.

Lastly, in order to compare the quantitative output of the model against the biological data, we counted the number of vertebral segments as well as the number of segmentation defects (small vertebrae or fusions) in mutant fish using *entpd5* expression at 28 dpf, when the vertebral bodies have developed into a mature form (*Figure 7—figure supplement 9A,B*). All mutants had a greater variability in segment number than wild type (n = 48), where segment number ranged from 29 to 31. In *her1*$^{-/-}$*;her7*$^{-/-}$ (n = 38) the segment number increased up to 37. The increase in *her1;her7* was slightly ameliorated by the additional removal of *tbx6* in the triple mutants (n = 47). As expected,

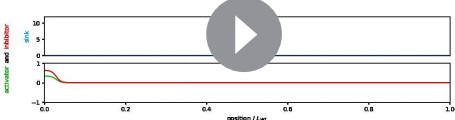

**Video 1.** Simulation of the autonomous sinkless condition. The absence of sink profile (blue) in the top panel and corresponding activator (green) and inhibitor (red) patterns in the bottom with patterning occurring sequentially from anterior to posterior. Parameters as in *Figure 7* of the main text.

DOI: https://doi.org/10.7554/eLife.33843.021

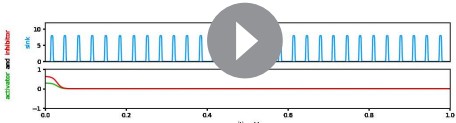

**Video 2.** Simulation of the wild type condition. The sink profile (blue) for the wild type condition in the top panel and corresponding activator (green) and inhibitor (red) patterns in the bottom. Parameters as in *Figure 7* of the main text.

DOI: https://doi.org/10.7554/eLife.33843.022

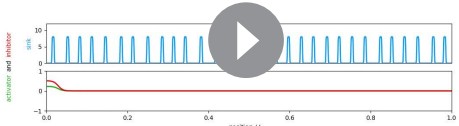

**Video 3.** Simulation of the *her1;her7* mutant condition. The sink profile (blue) showing the noisy spatial distribution for the *her1;her7* mutant condition in the top panel and corresponding activator (green) and inhibitor (red) patterns in the bottom. Parameters as in *Figure 7* of the main text.
DOI: https://doi.org/10.7554/eLife.33843.023

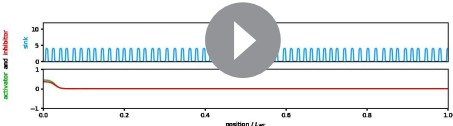

**Video 4.** Simulation of the *tbx6* mutant condition. The sink profile (blue) representing the noisy spatial distribution and the reduced amplitude of sinks in the *tbx6* mutant in the top panel and corresponding activator (green) and inhibitor (red) patterns in the bottom. This simulation also represents the *her1;her7; tbx6* mutant. Parameters as in *Figure 7* of the main text.
DOI: https://doi.org/10.7554/eLife.33843.024

*hes6* mutants (n = 41) had a lower number of segments than the wild type, ranging from 28 to 31 segments. When the same fish were scored for segmentation defects (*Figure 7—figure supplement 9B*), *her1*$^{-/-}$;*her7*$^{-/-}$ presented a higher frequency and a broader distribution of defects than *tbx6* mutants (n = 46). In *her1;her7;tbx6* mutants, the number of defects per embryo decreased almost to the distribution seen in *tbx6* mutants. The trends in these measurements are captured by the theory, as shown by the number of peaks in the pattern and (*Figure 7—figure supplement 9C*) the number of defects (*Figure 7—figure supplement 9D*), defined as segments of lengths outside the wild type segment length range.

## Discussion

In order to dissect the interrelationship of paraxial mesoderm segmentation and chordacentrum formation in teleosts, we here use the zebrafish to manipulate the segmentation clock in the paraxial mesoderm, while using a novel marker for axial segmentation to examine chordacentrum and vertebral body patterning at early and late stages.

The zebrafish notochord consists of a core of vacuolated cells responsible for turgor pressure, and an outer, squamous epithelial layer, the sheath cells, that is adjacent to the notochord's basal lamina or sheath, which provides mechanical stability (*Melby et al., 1996*; *Yamamoto et al., 2010*) (*Figure 3E*). Previously, we described the enzyme ectonucleoside triphosphate/diphosphohydrolase 5 (Entpd5) as essential for ossification in zebrafish (*Huitema et al., 2012*). *Entpd5* mutants fail to mineralize osteoid, leading to a complete absence of bone. *entpd5* is co-expressed with *osterix* in all osteoblasts of the early head skeleton, the cleithrum and the operculum (*Huitema et al., 2012*), but we here show that *entpd5* is also expressed in notochord cells (*Figure 3*), and that, at 4 dpf, the expression in the axial mesoderm becomes restricted to segmentally organized rings in the notochord sheath cells. *osx* is not expressed at these early stages in the notochord sheath cells, nor do

*osx* mutant present with an axial mineralization phenotype in zebrafish or Medaka (*Figure 3—figure supplement 1*; *Yu et al., 2017*). Combined, these results identify the *osx*-, *entpd5*+ sheath cells as those osteoblasts responsible for initial chordacentra formation, and indicate that the segmental patterning of the notochord is under way by 4 dpf in these cells. Mechanistically, the segmental expression of Entpd5 in the notochord sheath provides the enzymatic activity to mineralize the fibrous sheath of the notochord. It is possible that the respective contribution of notochord sheath cells in chordacentrum formation versus sclerotome-derived osteoblast function in centrum formation is different in other

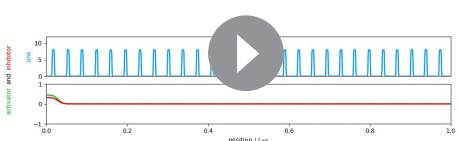

**Video 5.** Simulation of the *hes6* mutant condition. The sink profile (blue) representing the longer spatial wavelength of sinks in the *hes6* mutant in the top panel and corresponding activator (green) and inhibitor (red) patterns in the bottom. Parameters as in *Figure 7* of the main text.
DOI: https://doi.org/10.7554/eLife.33843.025

teleost species (*Fleming et al., 2015*; *Yasutake et al., 2004*; *Kaneko et al., 2016*), and this can be tested by generating *entpd5* mutants in some of those species. In zebrafish, our findings settle a long-standing debate on which cells are functionally important for chordacentrum formation.

From an evolutionary perspective, it has been suggested that the chordacentrum is a novelty found only in actinopterygians (*Arratia et al., 2001*). It seems safe to state that chordacentrum formation is conserved at least among teleosts, while the situation for holosteans (Amia and the gars) is less clear. Whether chondrosteans such as Acipenser, which lack centra but exhibit mineralized neural and hemal arches are different deserves further studies in order to bridge the gap between paleontological, anatomical and molecular studies (*Arratia et al., 2001*; *Laerm, 1976*; *Laerm, 1979*). Recent work has shown that clock-based mechanisms are involved in body segmentation in vertebrates as well as invertebrates (*Sarrazin et al., 2012*) and there is wide agreement that amniotes segmentally pattern their muscles and axial skeletons at the same time during development using the 'segmentation clock'. However, from fossil and extant taxa it seems likely that neural and hemal arches are evolutionarily older than vertebral bodies (*Cote et al., 2002*), and while these structures are homologous across vertebrate classes (*Williams, 1959*), vertebral bodies are not. The demonstration of periodic *entpd5* expression in the notochord sheath cells now provides a tool to study the earliest visible patterning events in the axial skeleton, and also allows an evaluation of the segmentation clock's input. To perturb the clock, we generated new alleles and new allelic combinations in *tbx6* and the central clock genes, *her1*, *hes6*, and *her7*. Analysis using a number of different readouts demonstrate that the phenotype of *her1;her7* double mutants is not exacerbated in the further absence of *hes6*. Strongest disruption of the clock is seen in the case of *her1;her7;tbx6* triple mutants. The mutants we have used to interfere with the segmentation clock and its output leave the PSM without any overt signs of oscillation or segmental pattern. It is of course formally possible that some remaining clock-like activity exists that we cannot detect. However, our argument in this paper does not require that we have removed all segmentation clock activity, but rather that our perturbations of the segmentation clock and its output show highly ordered axial structures despite strong disorder in the early paraxial mesoderm.

An influence of the paraxial mesoderm on axial segmentation may be reflected in the differences between the mutants' distributions of defects. The strongest axial phenotype is found in $her1^{-/-}$; $her7^{-/-}$, which possesses prominent myotome boundary fragments in the paraxial mesoderm, whereas $tbx6^{-/-}$ and $her1^{-/-};her7^{-/-};tbx6^{-/-}$ have a slightly weaker axial phenotype, which is preceded by a greater reduction of myotome boundaries. This is consistent with a dominant interfering effect of paraxial structures. The similarities in paraxial and axial phenotypes of $tbx6^{-/-}$ and $her1^{-/-}$; $her7^{-/-};tbx6^{-/-}$ is explained by proposing that the effects of a disrupted segmentation clock (that can nevertheless output a disrupted pattern seen in $her1^{-/-};her7^{-/-}$) are reduced in $her1^{-/-}$; $her7^{-/-};tbx6^{-/-}$ by removing the output function of *tbx6* in the anterior PSM.

We have attributed the occasional defects in the mutant centra to the effects of fluctuations in the strength and position of sinks for the inhibitor component of the notochord's intrinsic segmentation mechanism. In the absence of any sinks, the notochord would in our model segment without any defects with a periodicity determined entirely by the internal dynamics. Thus, the defects can be viewed as a 'dominant interference' by the mis-patterned paraxial mesoderm. Unfortunately, due to the random and unpredictable position of the chordacentra defects seen in all mutants analyzed, cell transplantation experiments cannot serve as a suitable experimental strategy to test this. We have also contemplated the possibility that *her1*, *her7* and *tbx6* are also expressed in the notochord and may be directly responsible for the centra defects seen in the mutants. However, this appears highly unlikely, because no *tbx6* (*Wanglar et al., 2014*), *mespb or her1* expression (In situ hybridization or transgene expression at 5 dpf and 13 dpf, data not shown) has been detected in the notochord at somite stages or during chordacentra formation. A recent report indicates that overexpression of *mespbb* in the notochord sheath can regulate segment size, possibly via interfering with boundary formation (*Wopat et al., 2018*).

Another interpretation is possible in which the notochord's intrinsic mechanism is highly noisy, and the paraxial mesoderm has no influence. While any molecular patterning mechanism would be vulnerable to fluctuations in the embryo, since the notochord's segments are generated over a long time scale, with a period about 20 times slower than somitogenesis, it seems plausible that the large molecular numbers that could be synthesized in this interval could ensure a low error rate. Furthermore, the transfer of information from a segmentation clock-derived muscle pre-pattern to an

autonomous but plastic mechanism in the notochord resolves the apparent discrepancy between *tbx6* and *hes6* mutant phenotypes, and explains how the animal coordinates segmented muscles and axial skeleton in biomechanical register despite the fact that they develop days to weeks apart in time. Such a coordination mechanism may be a widespread feature of the teleosts and potentially other vertebrates.

In summary, we have proposed a second mechanism for periodic segmentation of the axial skeleton of a vertebrate species, which exists in the sheath cells of the zebrafish notochord. The existence of intercalary defects argues that although there is a sequential generation of periodic *entpd5* expression rings, this periodicity does not arise from a clock-type segmentation mechanism as found in the paraxial mesoderm. We have described this dynamic mechanism as a reaction-diffusion process that sweeps down the axis, generating segments autonomously and refining their position using information from the previously segmented myotomes. The genetic basis of the proposed autonomous notochord mechanism and the signal from myotome to notochord are not understood, and our model does not assume any specifics with regard to molecular identities, but it opens the door to the search for these molecules. *entpd5* is the earliest known molecular marker for the sites of ossification, with plasticity in the presence of perturbation, suggesting that understanding the control of *entpd5* expression may hold the key.

# Materials and methods

**Key resources table**

| Reagent type or resource | Designation | Source or reference | Identifiers | Additional information |
|---|---|---|---|---|
| Gene (*Danio rerio*) | *sagff214* | NA | | |
| Gene (*Danio rerio*) | *entpd5* | NA | | |
| Gene (*Danio rerio*) | *her1* | NA | | |
| Gene (Danio rerio) | *her7* | NA | | |
| Gene (*Danio rerio*) | *tbx6* | NA | | |
| Gene (*Danio rerio*) | *hes6* | NA | | |
| Gene (*Danio rerio*) | *deltaD* | NA | | |
| Gene (*Danio rerio*) | *deltaC* | NA | | |
| Gene (*Danio rerio*) | *osterix* | NA | | |
| Genetic reagent | | | | |
| Genetic reagent (*Danio rerio*) | Tg(entpd5:kaede) | *Geurtzen et al., 2014* doi: 10.1242/dev.105817 | hu6867 | Same BAC used as *Huitema et al., 2012* with kaede insertion at first translated ATG |
| Genetic reagent (*Danio rerio*) | Tg(entpd5:pkred) | This paper | hu7478 | Same BAC used as *Huitema et al., 2012* with pkred insertion at first translated ATG |
| Genetic reagent (*Danio rerio*) | Tg(SAGFF214:GFP) | *Yamamoto et al., 2010* DOI: 10.1242/dev.051011 | | |
| Genetic reagent (*Danio rerio*) | Osterix:mcherry | *Spoorendonk et al., 2008* DOI: 10.1242/dev.024034 | hu4008 | |
| Genetic reagent (*Danio rerio*) | *her1* | *Schröter et al., 2012* doi: 10.1371/journal.pbio.1001364 | hu2124 | |
| Genetic reagent (*Danio rerio*) | *her7* | *Schröter et al., 2012* doi:10.1371/journal.pbio.1001364 | hu2526 | |
| Genetic reagent (*Danio rerio*) | *tbx6* | *Busch-Nentwich et al., 2013* ZFIN ID: ZDB-PUB-130425–4 | sa38869 | |
| Genetic reagent (*Danio rerio*) | *hes6* | *Schröter and Oates, 2010* doi: 10.1016/j.cub.2010.05.071 | zm00012575Tg | Also called zf288Tg |

*Continued on next page*

*Continued*

| Reagent type or resource | Designation | Source or reference | Identifiers | Additional information |
|---|---|---|---|---|
| Genetic reagent (Danio rerio) | *deltaD* | **van Eeden et al., 1996** PMID: 9007237 | ar33 | Also called tr233 |
| Genetic reagent (Danio rerio) | *deltaC* | **van Eeden et al., 1996** PMID: 9007237 | tm98 | |
| Recombinant DNA reagent (plasmid) | | | | |
| Plasmid (*Danio rerio*) | her7 | **Oates and Ho, 2002** | | |
| Plasmid (*Danio rerio*) | mespb | **Sawada et al., 2000** | | |
| Plasmid (*Danio rerio*) | xirp2a | **Deniziak et al., 2007** | | |
| Plasmid (*Danio rerio*) | papc | **Yamamoto et al., 1998** | | |
| Plasmid (*Danio rerio*) | en2a | **Erickson et al., 2007** | | |
| Plasmid (*Danio rerio*) | ripply1 | PCR template: Rip1 F (CGTGGCTTGTGACCAGAAAAG) Rip1 R T7 325 (TAATACGACTCACTATAGGCT GTGAAGTGACTGTTGTGT) | | |
| Plasmid (*Danio rerio*) | ripply2 | PCR template: Rip2 F(ACGCGAATCAACCCTGGAGA) and Rip2 R T7 281 (AATACGACTCACTATAGGGAGA GAGCTCTTTCTCGTCCTCTTCAT) | | |
| Plasmid (*Danio rerio*) | dlc | **Oates and Ho, 2002** | | |
| Plasmid (*Danio rerio*) | her1 | **Müller et al., 1996** | | |
| Sequence-based reagent | | | | |
| Talen | hes6 | this paper | | See *Figure 1*, *Figure 1—figure supplement 1* |
| Talen | her1 | this paper | | See *Figure 1*, *Figure 1—figure supplement 1* |
| Commercial assay or kit | | | | |
| Commercial assay or kit | RNeasy MinElute Cleanup Kit | Qiagen | Cat No./ID: 74204 | |
| Commercial assay or kit | Gene jet plasmid (miniprep kit) | Thermo scientific | Cat no: K0502 | |
| Chemical compound, drug | | | | |
| Chemical compound, drug | Alizarin red | Sigma | CAS Number 130-22-3 | |
| Software, algorithm | | | | |
| Software, algorithm | LAS X | Leica microsystems | | |
| Software, algorithm | Fiji (RRID:SCR_002285) | ImageJ 1.51 n | | |
| Software, algorithm | Python (RRID:SCR_008394) | Version Python 2.7.14: : Anaconda custom (64-bit) | | |
| Software, algorithm | Libraries: numpy (RRID:SCR_008633), matplotlib (RRID:SCR_008624) | Anaconda distribution | | |
| Software, algorithm | Lleras_fhn_1d_ solve_and_animate_ eLife.py | This paper | Custom PDE solver and animator | Provided as supplementary data. |
| Software, algorithm | Spyder | Anaconda distribution, Spyder 3.2.6 | The Scientific PYthon Development EnviRonment | |

## Animal procedures

All zebrafish strains were maintained at the Hubrecht Institute, the Institute of Cardiovascular Organogenesis and Regeneration, and at University College London. Standard husbandry conditions applied. Animal experiments were approved by the Animal Experimentation Committee (DEC) of the Royal Netherlands Academy of Arts and Sciences and by the UK Home Office under PPL 70/7675. Embryos were kept in E3 embryo medium (5 mM NaCl, 0.17 mM KCl, 0.33 mM CaCl$_2$, 0.33 mM MgSO$_4$) at 28°C. For anesthesia, a 0.2% solution of 3-aminobenzoic acid ethyl ester (Sigma), containing Tris buffer, pH 7, was used.

## Zebrafish lines

New transgenic lines (*entpd5:pkRED, entpd5:kaede*) were generated as described previously (*Huitema et al., 2012*). Fluorophores were recombined into the ATG site of the *entpd5* gene (BAC clone CH211-202H12). *fss*$^{sa38869}$, *bea*$^{tm98}$, *aei*$^{ar33}$, *her1*$^{hu2124}$ and *her7*$^{hu2526}$ mutants were acquired from Prof. Jeroen den Hertog. The Sagff214:galFF line was obtain from K. Kawakami. Double mutants for *her1* (in the *her7*$^{hu2526}$ background) and mutants for *hes6* were generated by TALEN injection (*Dahlem et al., 2012*) (see *Figure 1—figure supplement 1* for details). An *Osterix* mutant was created by Tilling (*Apschner, 2014*). The newly generated *hes6* mutant allele has one or two somites fewer (15 or 16 somites from anterior to the proctodeum, n = 78) compared to wild types (17 somites n = 15), consistent with the previously reported *hes6* mutant (*Schröter and Oates, 2010*).

## Genotyping

DNA was isolated through fin clippings and from embryos. Genotyping was performed as described in *Table 1* and *Table 2*.

## In situ hybridization and alizarin Red bone staining

Riboprobes were generated from either plasmids or PCR templates and in situ hybridisation was performed as previously described (*Oates and Ho, 2002*). Whole mount stained embryos were documented on an Olympus SZX10 stereoscope with a QImaging Micropublisher camera. Flat-mounted embryos were photographed on an Olympus MVX10 stereoscope with an Olympus DP22 camera. Alizarin Red bone staining was performed as described previously (*Spoorendonk et al., 2008*) for fish between 8 weeks and one year of age.

## Virtual time lapses

Embryos were kept in E3 at 28°C until 7 dpf. At 7 dpf, ten embryos with fully developed swim bladder from mutant (*tbx6*$^{-/-}$, *her1*$^{-/-}$;*her7*$^{-/-}$, *her1*$^{-/-}$;*her7*$^{-/-}$;*tbx6*$^{-/-}$, or *hes6*$^{-/-}$) or transgenic *entpd5: kaede* lines were anaesthetized, photographed and then housed individually in the animal facility. Individuals were fed tetrahymena in combination with Gemma 75 for the first two weeks, followed by artemia and Gemma 150 for the following two weeks. Every second day, each individual was, anesthetized (described above) and photographed using a Olympus SXZ16 stereomicroscope (1.5X PlanApo objective) connected to a DFC450C Leica camera. The embryo was placed in a drop of E3 with anesthetic on the lid of a petri dish. Each picture was taken from the cleithrum to the posterior tip of the individual. Immediately afterwards, the embryo was returned to warm E3

**Table 1.** Genotyping of lines using sequencing or restriction enzyme digestion

| Zebrafish line | FW | RV | Restriction enzymes |
| --- | --- | --- | --- |
| *her1* | TCTAGCAAGGACACGCATGA | GATGAAGAGGAGTCGGTGGA | |
| *her7* | GATGAAAATCCTGGCACAGACT | TCTGAATGCAGCTCTGCTCG | |
| *hes6* | TCACGACGAGGATTATTACGG | GGGCGACAACGTAGCGTA | NHEI |
| *her1*$^{-/-}$;*her7*$^{-/-}$ and *her1*$^{-/-}$;*her7*$^{-/-}$;*tbx6*$^{-/-}$ | ACTCCAAAAATGGCAAGTCG | GCCAATTCCAGAATTTCAGC | AGEI |
| *aei* | AGGGAAGCTACACCTGCTCA | TTCTCACAGTTGAATCCAGCA | |
| *fss* | GGGTCATTGTTGGGGTTGCA | ATGAACACCGCCCTTCCAAT | |

DOI: https://doi.org/10.7554/eLife.33843.036

**Table 2.** Genotyping using Kaspar

| Zebrafish line | FW X | FW Y | RV |
|---|---|---|---|
| *bea* | GAAGGTGACCAAGTTCATGCT | GAAGGTCGGAGTCAACGGATT | AGTCCTTGCCTGACAAACCAA |

DOI: https://doi.org/10.7554/eLife.33843.037

without anesthetic. Embryos were returned to their specific tank in the animal facility only when they were completely awake and moving. This procedure was repeated until all *entpd5+* segments had developed in the axis. The sedation and photography did not take more than two minutes per embryo, and did not compromise survival.

## Imaging

To photograph somite boundaries, 18–19.5 hpf embryos were dechorionated and laterally aligned in conical depressions that fit the yolk, in an agarose pad (Sigma, 2% in E3) cast in a petri dish (Falcon, 50 mm x 9 mm) and topped up with E3. Photomicrographs were taken on an Olympus MVX10 microscope equipped with an Olympus DP22 camera.

For imaging the notochord, embryos were mounted in 0.5% low melting point agarose in a culture dish with a cover slip replacing the bottom. Fluorescent imaging was performed with a Leica SPE 'live' Confocal Microscope, Leica SP8 confocal microscope and a PerkinElmer Ultraview VoX spinning disk microscope using a 10x or 20x objective with digital zoom. Usually, z-stacks with intervals of approximately 2 µm were captured and were then flattened by maximum projection in ImageJ. For photoconversion of whole *entpd5*:kaede embryos, the green Kaede fluorophore was photoconverted using a Leica fluorescence microscope by 15–30 min exposure through a UV light bandpass filter (360/40 nm, 100 W mercury lamp). For bone stains an Olympus S2 × 16 microscope coupled to a Leica DFC420c camera was used. Photomicrographs were stitched with the pairwise stitching plugin in Fiji (RRID:SCR_002285) (*Preibisch et al., 2009*).

## Theory

The theory describes notochord sheath cells pattern formation in terms of a one dimensional reaction diffusion system with two components, an activator U and an inhibitor V, see *Figure 7—figure supplement 1A*. The concentration $U = [U]$ of the activator is reflected in Entpd5 concentration. The concentrations of both the activator and inhibitor species $U(x,t)$ and $V(x,t)$ depend on position $x$ and time $t$. We propose a variant of the FitzHugh-Nagumo (FHN) model (*Murray, 1993*)

$$\frac{\partial U}{\partial t} = D_U \frac{\partial^2 U}{\partial x^2} + k_1 U - k_3 U^3 - k_4 V + k_0 \tag{1}$$

$$\frac{\partial V}{\partial t} = D_V \frac{\partial^2 V}{\partial x^2} + k_5 U - k_6 V \tag{2}$$

where $D_U$ and $D_V$ are diffusion coefficients for $U$ and $V$ respectively, and $k_i$ are rate constants. The choice of the FHN model is based on its simplicity. There are positive linear terms for the activator and negative linear terms for the inhibitor in both *Equations (1) and (2)*, and a single nonlinearity, the cubic term for the activator that limits growth and allows for the stabilization of steady states. The model is meant to represent a plausible mechanism rather than specifying the interactions of particular molecules.

We reduce the number of parameters by transforming this theory to a dimensionless form. We first set the source term $k_0 = 0$ since as discussed below we need to reproduce a sequential pattern formation. We introduce a lengthscale $L$ that we will take as the system size, and a timescale $T$ and concentration scale $U_0$ to be set below. In terms of these scales we define new variables $x'$, $t'$, $u$ and $v$ such that

$$x = L x' \tag{3}$$

$$t = T t' \tag{4}$$

$$U = U_0 u \tag{5}$$

$$V = U_0 v \tag{6}$$

and replace in the reaction diffusion equations above dropping the primes for notational convenience

$$\frac{U_0}{T}\frac{\partial u}{\partial t} = \frac{D_U U_0}{L^2}\frac{\partial^2 u}{\partial x^2} + k_1 U_0 u - k_3 U_0^3 u^3 - k_4 U_0 v \tag{7}$$

$$\frac{U_0}{T}\frac{\partial v}{\partial t} = \frac{D_V U_0}{L^2}\frac{\partial^2 v}{\partial x^2} + k_5 U_0 u - k_6 U_0 v. \tag{8}$$

We multiply both equations by $T/U_0$ to render them dimensionless

$$\frac{\partial u}{\partial t} = \frac{D_U T}{L^2}\frac{\partial^2 u}{\partial x^2} + k_1 T u - k_3 T U_0^2 u^3 - k_4 T v \tag{9}$$

$$\frac{\partial v}{\partial t} = \frac{D_V T}{L^2}\frac{\partial^2 v}{\partial x^2} + k_5 T u - k_6 T v. \tag{10}$$

Multiplying the inhibitor equation by $k_1/k_5$ and rearranging terms

$$\frac{\partial u}{\partial t} = \frac{D_U}{k_1 L^2}(k_1 T)\frac{\partial^2 u}{\partial x^2} + (k_1 T)u - \frac{k_3 U_0^2}{k_1}(k_1 T)u^3 - \frac{k_4}{k_1}(k_1 T)v \tag{11}$$

$$\frac{k_1}{k_5}\frac{\partial v}{\partial t} = \frac{D_V}{k_5 L^2}(k_1 T)\frac{\partial^2 v}{\partial x^2} + (k_1 T)u - \frac{k_6}{k_5}(k_1 T)v \tag{12}$$

where we have highlighted dimensionless groups in parentheses. We now select a timescale by setting

$$k_1 T \equiv 1 \tag{13}$$

and a concentration scale by setting

$$\frac{k_3 U_0^2}{k_1} \equiv 1 \tag{14}$$

and define dimensionless parameter groups

$$\tau \equiv \frac{k_1}{k_5}, a \equiv \frac{D_U}{k_1 L^2}, b \equiv \frac{D_V}{k_5 L^2}, \kappa_4 \equiv \frac{k_4}{k_1}, \kappa_6 \equiv \frac{k_6}{k_5}. \tag{15}$$

With these definitions

$$\frac{\partial u}{\partial t} = a\frac{\partial^2 u}{\partial x^2} + u - u^3 - \kappa_4 v \tag{16}$$

$$\tau\frac{\partial v}{\partial t} = b\frac{\partial^2 v}{\partial x^2} + u - \kappa_6 v. \tag{17}$$

We additionally set $\kappa_4 = 1$ for simplicity and we call $\kappa_6 = d$

$$\frac{\partial u}{\partial t} = a\frac{\partial^2 u}{\partial x^2} + u - u^3 - v \tag{18}$$

$$\tau\frac{\partial v}{\partial t} = b\frac{\partial^2 v}{\partial x^2} + u - dv. \tag{19}$$

In the rest of this work we consider this dimensionless form of the theory. Here $a$ and $b$ are dimensionless scaled diffusion coefficients of the activator and inhibitor species respectively, $\tau$ is a relative timescale, and $d$ is a dimensionless degradation constant of the inhibitor.

We first consider the homogeneous system $\partial_{xx}u = \partial_{xx}v = 0$. The resulting equations for the local reactions are

$$\dot{u} = u - u^3 - v \qquad (20)$$

$$\tau\dot{v} = u - dv. \qquad (21)$$

where dots denote time derivatives. Introducing functions

$$f(u,v) = u - u^3 - v \qquad (22)$$

$$g(u,v) = \tau^{-1}u - \tau^{-1}dv, \qquad (23)$$

the nullclines of the system, defined by setting $f(u,v) = 0$ and $g(u,v) = 0$, are the curves in the $(u,v)$ plane

$$v = u - u^3, \qquad (24)$$

$$v = d^{-1}u. \qquad (25)$$

The first one is an inverted cubic that goes through the origin and the second one is a linear function with slope $d^{-1}$ controlled by the single bifurcation parameter $d$, see *Figure 7—figure supplement 1B*. Intersections of these two curves are the solutions to

$$u\left(u^2 + d^{-1} - 1\right) = 0 \qquad (26)$$

and define the fixed points of the system where $\dot{u} = \dot{v} = 0$. There is always a solution $(u_0, v_0) = (0,0)$ and for $d>1$ there are two additional solutions $u_\pm$ satisfying

$$u_\pm^2 = d^{-1} - 1. \qquad (27)$$

The linear stability of fixed points is determined by the matrix

$$A = \begin{pmatrix} f_u & f_v \\ g_u & g_v \end{pmatrix} \qquad (28)$$

where

$$f_u = \partial_u f(u,v) \qquad (29)$$

$$f_v = \partial_v f(u,v) \qquad (30)$$

$$g_u = \partial_u g(u,v) \qquad (31)$$

$$g_v = \partial_v g(u,v) \qquad (32)$$

and derivatives are evaluated at the fixed point $(u_*, v_*)$. The condition for stability is that

$$\det A = f_u g_v - f_v g_u > 0 \qquad (33)$$

and

$$\operatorname{tr} A = f_u + g_v < 0. \qquad (34)$$

For the fixed point $(u_0, v_0) = (0,0)$ we obtain

$$A = \begin{pmatrix} 1 & -1 \\ \tau^{-1} & -d\tau^{-1} \end{pmatrix} \tag{35}$$

with determinant and trace

$$\det A = (1-d)\tau^{-1} \tag{36}$$

$$\operatorname{tr} A = 1 - d\tau^{-1}. \tag{37}$$

Given that $\tau, d > 0$ this implies that the origin $(0,0)$ is a stable fixed point if

$$d < 1 \quad \text{and} \quad \tau < d. \tag{38}$$

We consider in the following a situation in which the fixed point $(0,0)$ is stable, setting the dimensionless timescale $\tau = 0.1$ and degradation $d = 0.5$. Under appropriate conditions, the dimensionless reaction diffusion theory *Equations (18) and (19)* can give rise to pattern formation. In particular we require that the activator diffuses slower than the inhibitor, $a < b$, and here set $a = 10^{-3}$ and $b = 10^{-2}$. Because of the signs of the derivatives near the fixed point, the type of pattern predicted is as displayed in *Figure 7—figure supplement 1C*, where there is coexistence of activator and inhibitor (*Murray, 1993*).

In the presence of random perturbations distributed along the notochord the homogeneous state loses stability due to differential diffusion. A pattern may form out of this initial random background fluctuation, with segments forming almost simultaneously all along the notochord. Although segment formation is robust in this scenario and can accommodate a broad range of wavelengths, this is at odds with the experimental observation that ENTPD5 segments form sequentially from anterior to posterior.

We conjectured that one way to obtain a sequential segment formation is to start with an initial perturbation localized at the anterior, and vanishing concentrations all across the rest of the notochord. Since the anterior of the vertebrate axis is always more developmentally advanced than the posterior, such an anterior perturbation is a plausible hypothesis. A vanishing concentration along the notochord is important to ensure that patterning is not triggered until the wave of activator and inhibitor arrives at a given point. Therefore, we start simulations with initial conditions that have zero concentration for both $u$ and $v$ across the whole domain $x \in (0, L)$, except for a small perturbation near the origin $x = 0$. The form of the initial condition is a smooth step

$$u(x,0) = \frac{u_0}{2}(1 - \tanh(5.0(x - 0.5))) \tag{39}$$

$$v(x,0) = \frac{v_0}{2}(1 - \tanh(5.0(x - 0.5))) \tag{40}$$

with a steepness 5.0 and width 0.5. Initial values $u_0$ and $v_0$ are determined randomly from a uniform distribution in the interval $(0.1, 0.2)$, see examples in *Figure 7—figure supplements 1–5*. Starting from such a small perturbation in the anterior, we observe that the system is able to form a pattern sequentially, from anterior to posterior, see *Figure 7A*, *Figure 7—figure supplement 2* and *Video 1*. Thus, the theory proposed is capable of autonomous patterning of the notochord in the absence of input from the segmentation clock.

We next introduce the effect of the segmentation clock input into this otherwise autonomous patterning system as a spatial dependent degradation profile of the inhibitor

$$\frac{\partial u}{\partial t} = a\frac{\partial^2 u}{\partial x^2} + u - u^3 - v \tag{41}$$

$$\tau\frac{\partial v}{\partial t} = b\frac{\partial^2 v}{\partial x^2} + u - dv - s(x)v. \tag{42}$$

This *sink profile* $s = s(x)$ for the inhibitor has peaks at given positions along the $x$ axis, describing the cues that the notochord patterning mechanism receives from myotomes. At positions where $s(x)$

is large, the inhibitor is locally degraded at a larger rate. Note that there is no source term for the activator since we set $k_0 = 0$ above. This feature together with the choice of sinks instead of sources to describe the segmentation clock cues are motivated from the observation that Entpd5 segments form sequentially. The presence of sources would render pattern formation non sequential.

The sink profile $s(x)$ is characterized by a sink strength $S_0$, a wavelength $\lambda$ and sink wavelength variability $\sigma$. The first sink is positioned at $\lambda/2$ and the positions $X_i$ of consecutive sinks are determined by the wavelength $\lambda$ with an error drawn from a uniform distribution of width $\sigma$. The sink profile is built from a combination of $\tanh(...)$ functions to produce smooth peaks of steepness $\alpha$ and width $\delta S$

$$s(x) = \frac{S_0}{2} \sum_i (-\tanh(\alpha(-X_i + x - \delta S)) + \tanh(\alpha(-X_i + x + \delta S))). \tag{43}$$

In this work we fix the values $\alpha = 100$ and $\delta S = 0.05$. The values of $S_0$, $\lambda$ and $\sigma$ are changed to describe the different conditions, see examples in *Figure 7* and *Figure 7—figure supplement 4*.

We consider a system size $L$ that we set to $L = 17.1$ so that the wildtype condition makes 30 segments with the sink profile natural wavelength $\lambda = 0.57$. We normalize axes length scales to this value in all plots. For simplicity we assume that the activator and inhibitor are restricted to notochord sheath cells and we specify Neumann boundary conditions, that is derivatives at both ends are zero

$$\frac{\partial u}{\partial x}\Big|_{x=0} = \frac{\partial v}{\partial x}\Big|_{x=L} = 0. \tag{44}$$

As described above, here we ensure the sequential character of the patterning through an initial perturbation at the anterior and vanishing concentrations across the notochord. One may query the robustness of such scenario, since noise across the notochord hampers sequential patterning. An alternative hypothesis would be to postulate an additional wavefront that propagates through the notochord progressively turning on the reaction diffusion mechanism of *Equations (41) and (42)* as it goes. To illustrate this we consider an alternative dimensionless form of *Equations (1) and (2)*. Turning back to *Equations (9) and (10)* we select a timescale and concentration scale setting

$$\frac{D_U T}{L^2} \equiv 1 \tag{45}$$

and

$$\frac{k_3 U_0^2}{k_1} \equiv 1. \tag{46}$$

Introducing dimensionless parameter groups

$$\delta \equiv \frac{D_V}{D_U}, \gamma \equiv \frac{k_1 L^2}{D_U}, \kappa_i \equiv \frac{k_i}{k_1}, \tag{47}$$

and setting for simplicity $\kappa_4 = \kappa_5 = 1$ and $\kappa_6 = \kappa$ we arrive at the dimensionless form

$$\frac{\partial u}{\partial t} = \frac{\partial^2 u}{\partial x^2} + \gamma(u - u^3 - v), \tag{48}$$

$$\frac{\partial v}{\partial t} = \delta \frac{\partial^2 v}{\partial x^2} + \gamma(u - \kappa v). \tag{49}$$

In this alternative dimensionless form it is straightforward to decouple the reactions from diffusion by tuning the value of $\gamma$. Thus, we can introduce a wavefront $\gamma(x, t)$ that moves from anterior to posterior turning on the reactions in its wake. Such a wavefront could have a biological origin in a molecular maturation gradient invading the notochord from the anterior. Due to very slow dynamics before wavefront arrival, this would render the patterning mechanism more robust to noise across the notochord. Yet a different possibility is a scenario of patterning in a growing domain (*Crampin et al., 1999*), although here the tissue where the pattern forms exists previous to the establishment of the pattern. These alternatives are certainly interesting and the underlying

mechanism patterning the notochord will be the subject of future work. However, in the absence of experimental data supporting other hypotheses, here we settle on the perhaps more parsimonious choice of vanishing initial conditions across the notochord.

In this work we solve the partial differential equations described above using a custom python code, see *Source code 1*. We discretize space with a discretization length $\Delta x = 0.01$, and time discretization is chosen as $\Delta t = 0.9\Delta x^2/2$. We integrate the partial differential equations until the pattern reaches a steady state.

## Acknowledgements

We thank the Schulte-Merker and Oates groups for constructive discussions. The Institute of Biostatistics and Clinical Research University of Münster provided helpful insights, the IT-Zentrum Forschung und Lehre (WWU Münster) rendered IT support. Numerous technicians and part time students helped in genotyping and maintaining zebrafish lines. Prof. J den Hertog provided the *her1*, *her7, tbx6, aei* and *bea* mutant lines to SSM, Prof. K Kawakami the Sagff214:galFF strain. This study was also funded by The Francis Crick Institute, receiving its core funding from Cancer Research UK, the Medical Research Council, and Wellcome to RN and ACO.

## Additional information

### Funding

| Funder | Grant reference number | Author |
|---|---|---|
| Deutsche Forschungsgemeinschaft | CIM1003 | Laura Lleras Forero Stefan Schulte-Merker |
| Medical Research Council | MC_UP_1202/3 | Rachna Narayanan Andrew C Oates |
| Francis Crick Institute | | Rachna Narayanan Andrew C Oates |
| Wellcome | WT098025MA | Guillaume Valentin Andrew C Oates |
| Fondo Para la Convergencia Estructural del MERCOSUR | COF 03/11 | Luis G Morelli |
| Agencia Nacional de Promoción Científica y Tecnológica | PICT 2012 1954 | Luis G Morelli |
| Agencia Nacional de Promoción Científica y Tecnológica | PICT 2013 1301 | Luis G Morelli |

The funders had no role in study design, data collection and interpretation, or the decision to submit the work for publication.

### Author contributions

Laura Lleras Forero, Conceptualization, Formal analysis, Investigation, Visualization, Methodology, Writing—original draft, Writing—review and editing; Rachna Narayanan, Investigation, Writing—original draft, Writing—review and editing; Leonie FA Huitema, Maaike VanBergen, Alexander Apschner, Josi Peterson-Maduro, Ive Logister, Guillaume Valentin, Investigation; Luis G Morelli, Conceptualization, Data curation, Software, Funding acquisition, Validation, Writing—original draft, Writing—review and editing; Andrew C Oates, Conceptualization, Software, Formal analysis, Funding acquisition, Methodology, Writing—original draft, Writing—review and editing; Stefan Schulte-Merker, Conceptualization, Funding acquisition, Writing—original draft, Project administration, Writing—review and editing

## Author ORCIDs
Luis G Morelli (iD) http://orcid.org/0000-0001-5614-073X
Andrew C Oates (iD) http://orcid.org/0000-0002-3015-3978
Stefan Schulte-Merker (iD) http://orcid.org/0000-0003-3617-8807

## Ethics

Animal experimentation: Animal experiments were approved by the Animal Experimentation Committee (DEC) of the Royal Netherlands Academy of Arts and Sciences and by the UK Home Office under PPL 70/7675

## Decision letter and Author response

Decision letter https://doi.org/10.7554/eLife.33843.042
Author response https://doi.org/10.7554/eLife.33843.043

## Additional files

### Supplementary files
• Source code 1. Custom python code.
DOI: https://doi.org/10.7554/eLife.33843.038

• Transparent reporting form
DOI: https://doi.org/10.7554/eLife.33843.039

### Data availability

Virtual time lapse data and theory code can be found at: http://icor-data.uni-muenster.de/. Source data files and source code have been submitted to eLife as additional material.

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
