## [Decision Letter]

Thank you for submitting your article "Segmentation of the zebrafish axial skeleton relies on notochord sheath cells and not on the segmentation clock" for consideration by *eLife*. Your article has been reviewed by two peer reviewers, and the evaluation has been overseen by a Reviewing Editor (Tanya Whitfield) and Didier Stainier as the Senior Editor. The following individual involved in review of your submission has agreed to reveal his identity: Matthew Harris (Reviewer #1).

The reviewers have discussed the reviews with one another and the Reviewing Editor has drafted this decision to help you prepare a revised submission

Summary:

Lieras-Forero et al. detail quite novel and important findings on the independent patterning role of the notochord in segmentation of the zebrafish vertebral column in a process that is independent of somite formation. The wavefront model has been accepted as the conserved mechanism by which segmentation in vertebrate organisms occurs. It is/was generally assumed, however, that in fishes the meristic patterning is determined by a wavefront model as shown in beginning mutants in regulators of this patterning mechanism. The authors clearly, and elegantly, demonstrate underlying capacity of the zebrafish notochord to form ordered, meristic array of vertebral bodies even in the case of dysfunctioning segmental patterning in the sclerotome. These data build on classic comparative anatomy and recent genetic data that point to patterning of the chordacentra and the notochord as a key, and ancestral, step to the formation of vertebrae and patterning of the vertebral column. The authors further develop and test a mathematical model that is sufficient to explain the interaction between the dual patterning systems that can explain a number of characteristics seen in the adult phenotypes caused by different mutant combinations.

Essential revisions:

Both reviewers were positive overall, but each had suggestions for improvement. In particular, there are several concerns over the modelling aspect of the paper. Please address the following as essential revisions:

1) Please address all issues raised by both reviewers concerning the mathematical model.

2) Please address the issues concerning the genetic analysis and nomenclature raised by reviewer 1.

3) Please use the Discussion section to set your work in a broader context (see comments and suggestions from reviewer 1).

Reviewer #1:

The findings detailed by this paper are quite interesting and are important for developmental biologists broadly. The paper is very well written, and beyond some small (but essential) comments/critiques that I hope will be taken under consideration to increase the accuracy and impact of the manuscript, I believe it will be a landmark paper.

Genetics:

– Do new alleles generated/described in this manuscript fail to complement the previous described alleles? How do the authors know that these are nulls or severe loss-of-function? The position in of the TALEN alleles would support potential gene fragments to be produced. Similarly, what genetic characterization has been done on the Tilling alleles of her1 and her7 to see if these are true loss-of-function. Some discussion/exploration of characterization of these alleles in the paper would be helpful, however as the phenotype is what is critical in this manuscript, no extensive genetic analysis is necessary to be undertaken to address these questions.

– Giving triple mutants different mutant names is not commonly accepted by the field, nor is it helpful. Gollum, or bachau are not a single locus as the names would suggest. This is quite confusing and severely complicates conceptual understanding of comparisons between the compound mutants. This detracts from an otherwise exceptional and elegantly performed study.

Modeling

– The generation of the reaction diffusion model to integrate the notochordal and somatic patterning events is quite helpful and at least supplemental figure 14 should be in the paper. This is critical.

– The model shows only resting states. Please comment in the text if the model can reproduce the ontological sequence of patterning as shown in this paper (e.g. does it replicate a rostral bias?). Also, does the model cause fill-in responses to perturbations as seen in the mutant phenotypes.

– Subsection “Disruption of the segmentation clock in double and triple mutants” The authors may want to integrate the fact that the reaction diffusion mechanism provides plasticity by the feedback characteristics of the interactions.

Phylogenetic character analysis of notochord induction/association of the chordacentrum.

– At several points the authors detail current thinking of chordacentrum involvement in patterning of the vertebra column and formation of the centra. It would be important, and clarifying, if the authors discuss classical work in basal teleosts such as Amia and Gar suggesting that notochord induction/association of a chordacentrum is ancestral in teleosts (Schultze and Arratia papers in the 1980s) and addressed in Laern, (1976). How much of the discordance between models/species that the authors mention represents different mechanisms or, alternatively, similar mechanisms studied at different levels of analysis?

Reviewer #2:

In this study Forero et al., investigate the role of the segmentation clock in patterning chordacentra in zebrafish. Using a family of segmentation clock mutants, they disrupt segmentation to varying degrees and measure the chordacentra pattering. They propose a model in which the periodic patterning of chordacentra arises from a pattering process that can function independently from the segmentation clock. However, the somite patterns can modify the chordacentra patterning mechanism.

My major issue is that the model ought to generate periodic patterning of chordacentra in the absence of somitogenesis clock input. If the proposed model can do this robustly then some elementary exploration of this case is necessary (i.e. a minimum requirement is to provide enough detail so the results are reproducible and verify that there is a robust patterning mechanism in the case of no sinks). This validation is crucial as later the authors use the model to explain the location of activator peaks relative to sinks.

1) The authors claim that 'in the absence of any sinks, the notochord would in our model segment without any defects with a periodicity determined entirely by the internal dynamics.'

The key figure supporting this statement (Figure 14E) indicates that the mathematical model produces spatially periodic patterns at steady state in the absence of dermatome signal (i.e. s(x)=0 for all x).

It is not adequately explained in the text how the proposed model does this. What is the patterning mechanism?

A standard way to analyse the model is to consider behavior without diffusion. What are the steady states and what is their linear stability?

My analysis suggests that in the absence of diffusion and with s_0_<1-d (i.e. small influence from the dermatome) there is a unique steady state (0,0) that is stable.

When s_0_>1-d the origin becomes unstable and there are two non-zero steady states (i.e. presumably the model becomes bistable in this regime).

This analysis is consistent with the authors numerical results; the spatially homogeneous steady state is destabilized in simulations where the sink strength is greater than 0.5. Outside of this region the spatially homogeneous steady state is monostable.

Given the case where s_0_=0 is stable, the question then is whether the introduction of diffusion could cause an instability (e.g. Turing) and hence periodic patterning. If this is the case the authors should show it. However, even if this were true the wavefront behavior presented by the authors is nontrivial.

I have tried to reproduce Figure 14E with my own code and as many of the details provided but cannot.

2) The PDE model has a parameter that is discontinuous in space. The authors ought to provide details of their discretisation scheme so that the reader can assess how they have dealt with this discontinuity.

I suggest the following improvement: define the sinks independently of the numerical mesh and then approximate the steep switches with a continuous function such as tanh. In this way the sink strength parameter can be guaranteed to be continuous.

3) Stability analysis – is the pattering mechanism robust to small perturbations? It is worrying that the authors initialise on an unstable steady state of the homogeneous problem. Hence an infinitesimally small perturbation from these initial conditions could result in completely different model behaviour.

Could the authors add some small amplitude noise to the initial conditions and present some numerical results. Is the proposed wavefront solution stable?

4) The use of Fitzhugh Nagumo ought to be justified. I am not suggesting that the model needs to be linked to a molecular detail but some insight into the various terms would be helpful. The authors should describe the model assumptions and how they might be relevant to this system.

5) “In the simulations of fss, guu, and fum this strict correspondence between sink and activator is lost (Figure 7B and C); activator peaks occur both together with sinks and in between them.”

Can the authors use the model to provide insight into how this can happen? Is this observation a generic feature of activator-inhibitors models? If it is generic, then showing results from other reaction-diffusion models would help. If it is not generic, then the properties of the proposed model that yield the interesting behaviour ought to be defined and investigated more thoroughly.

6) Given the mathematical model takes up almost two pages of the results then I suggest that a figure exploring the relevant features of the model is appropriate.

[Editors' note: further revisions were requested prior to acceptance, as described below.]

Thank you for resubmitting your work entitled "Segmentation of the zebrafish axial skeleton relies on notochord sheath cells and not on the segmentation clock" for further consideration at *eLife*. Your revised article has been evaluated by Didier Stainier (Senior editor), Tanya Whitfield (Reviewing editor), and two reviewers.

The manuscript has been improved but there are some remaining issues that need to be addressed before acceptance. Reviewer 1 only has minor concerns that will be quick to address. Reviewer 2 has some more substantial concerns regarding robustness of the system to small perturbations. The reviewer has given comments together with two video files. Please address the comments from both reviewers.

Reviewer #1:

In the text the authors often list fss, her1;her7, and tbx6;her1;her7 mutant combinations. I assume the fss allele of tbx6 is the one being used (unless another has been generated). If this is the case the text should reflect this as tbx6-/-. her1; her7, tbx6;her1;her7 mutants. The authors have correctly labeled this in the figures, but not the figure legends nor text.

Somewhere in the text the authors should address whether these alleles are thought to be null or strong loss-of-function. Data is not needed, rather citation of previous genetic analysis on available alleles and a statement in the text is just helpful.

Results section “A reaction-diffusion model of axial patterning in the zebrafish”: “intrinsic segmentation mechanism, likely sheath cells" Sheath cells is not a mechanism. Do the authors mean within sheath cells?

Reviewer #2:

1) Whilst the theory section has been improved, now that it is explicit that the authors are proposing the Turing mechanism as the underlying patterning mechanism, can they provide a fuller analysis of the unstable wavenumbers for the parameter values presented in the simulations? It is important to characterize how the unstable wavenumbers (and corresponding wavelengths) relate to the typical inter-sink distance (e.g. presumably the model parameters have been chosen to give a wavenumber that is approximately of the same order as the inter-sink distance).

2) As the authors did not present the results with arbitrarily small noise in the initial data as I requested, I have solved the equations myself.

In Noise.mp4 I solve the model as it is presented in the paper. Note that the key qualitative behaviour is that independent of somite signal, a propagating wavefront leaves a periodic pattern in its wake. This is the behavior observed experimentally that any reasonable model ought to replicate.

In NoNoise.mp4 I have added very low amplitude noise to the initial conditions. Note that the noise destabilises the wavefront solution and the domain patterns simultaneously rather than sequentially.

These numerical results indicate that even the addition of infinitesimally small noise throughout the domain results in the wavefront solution being lost. Given the presence of noise in biological systems, this robustness issue is a fundamental limitation that, by not addressing in the main text of the manuscript, the authors seem to have neglected.

I suggest the following:

i) The authors build a convincing case, with reference to the pattern formation literature, that deals with the robustness issue. i.e. are there other published examples of wavefront propagation mediated spatial patterning with a similar lack or robustness?

or

ii) The authors build on their proposal that robustness could be mediated by a maturation gradient. This could be incorporated into the model by considering a competency domain where the authors solve the reaction diffusion equations on some domain [0, s(t)] where s(t) is a suitably chosen function of time.

Such a model would have a fundamentally different behaviour in that the imposed wavefront would determine the speed of segmentation rather than the Turing instability. Moreover, it would be robust to infinitesimally small noise.

For the suggested approach the authors could see, for example, Madzvamuse et al., 2005 or Crampin et al., 2002.

---

## [Author Response]

Essential revisions:Both reviewers were positive overall, but each had suggestions for improvement. In particular, there are several concerns over the modelling aspect of the paper. Please address the following as essential revisions:1) Please address all issues raised by both reviewers concerning the mathematical model.2) Please address the issues concerning the genetic analysis and nomenclature raised by reviewer 1.3) Please use the Discussion section to set your work in a broader context (see comments and suggestions from reviewer 1).Reviewer #1:The findings detailed by this paper are quite interesting and are important for developmental biologists broadly. The paper is very well written, and beyond some small (but essential) comments/critiques that I hope will be taken under consideration to increase the accuracy and impact of the manuscript, I believe it will be a landmark paper.

We thank the reviewer for the encouraging comments and the suggestions. We have implemented the requested changes and trust that the manuscript has improved.

Genetics:– Do new alleles generated/described in this manuscript fail to complement the previous described alleles? How do the authors know that these are nulls or severe loss-of-function? The position in of the TALEN alleles would support potential gene fragments to be produced. Similarly, what genetic characterization has been done on the Tilling alleles of her1 and her7 to see if these are true loss-of-function. Some discussion/exploration of characterization of these alleles in the paper would be helpful, however as the phenotype is what is critical in this manuscript, no extensive genetic analysis is necessary to be undertaken to address these questions.

The her1 talen allele has 2 consecutive premature stop codons (TAA TAA) and the her7 (hu2526) also has a premature stop codon (TAA). We cannot determine if *her1^-/-^;her7+/-; tbx6^-/-^* is a null on the protein level, because we don’t have antibodies to detect Her1 and Her7, but protein prediction suggests that the truncated proteins does not have wild type functionality. Her1 and Her7 are members of the bHLH superfamily of transcription factors – they need to dimerise for activity and they do this though the HLH domain and bind DNA with the basic (b) domain. The predicted Her1 protein product from the TALEN induced mutation lacks the basic DNA binding domain and the HLH dimerisation domains, so the prediction suggests that the protein product has no functionality. The predicted Her7 protein product of the hu2526 allele is also truncated because the stop codon is in the HLH domain – hence, this protein product is also predicted to be non-functional as its dimerising ability is lost. Importantly, the phenotype of *her1,her7* double mutantsis consistent with the phenotype of *her1,her7* double morphants and also of the b567 deletion allele (Henry et. al 2002). We did carry out a cross of *her1,her7* to the single *her1* mutant, and found that the *her1^-/-^;her7^+/-^* genotype presents a exacerbated phenotype than *her1* single mutants.

As describe in the Materials and methods section the Hes6 mutant generated here has the same phenotype as describe in the morpholino injections from Schroter and Oates, 2010.

– Giving triple mutants different mutant names is not commonly accepted by the field, nor is it helpful. Gollum, or bachau are not a single locus as the names would suggest. This is quite confusing and severely complicates conceptual understanding of comparisons between the compound mutants. This detracts from an otherwise exceptional and elegantly performed study.

We thank the reviewers for this comment. Indeed, we had discussed that among ourselves, and had prior to submission asked for input from the ZF nomenclature committee. While we believe the suggested names would have helped readability, we have now removed all names for the triple and double mutants, and now refer to all mutants by the genotype.

Modeling– The generation of the reaction diffusion model to integrate the notochordal and somatic patterning events is quite helpful and at least supplemental figure 14 should be in the paper. This is critical.

We thank the reviewer for this suggestion. We have moved previous Figure S14 into the main text in the revised manuscript (now Figure 7). Please note that we have also included in this main text figure now a new panel A showing the behavior of the model in the absence of sinks.

– The model shows only resting states. Please comment in the text if the model can reproduce the ontological sequence of patterning as shown in this paper (e.g. does it replicate a rostral bias?). Also, does the model cause fill-in responses to perturbations as seen in the mutant phenotypes.

We provided movies of the numerical simulations of the theory for the different conditions described in main text Figure 7. We also added new figure supplements (Figure 7—figure supplement 1 to Figure 7—figure supplement 5) which show a sequence of snapshots of these movies at several time points.

Both the movies and the snapshots illustrate that the model can reproduce the sequential dynamics of segment formation, from rostral to caudal. We now highlight this aspect of the theory in the main text.

As for fill-in responses observed in experiments, the data we have produced with the model is not conclusive. It may seem from the movies that when a defect is formed in place of a normal segment, the speed at which the activator concentration grows is slower than that of a normal segment. If there is a secondary mechanism reading out this slower and weaker increase, it may be that this could result in a quicker formation of the follow up segment and later formation of the defective one. This could describe the fill-in responses observed in experimental data. However, we have not been able to quantify this.

– Subsection “Disruption of the segmentation clock in double and triple mutants” The authors may want to integrate the fact that the reaction diffusion mechanisms provide plasticity by the feedback characteristics of the interactions.

We thank the reviewer for this suggestion. We now highlight the fact that the reaction diffusion mechanism can support a range of wavelengths and argue how this plasticity may allow the bias introduced by the segmentation clock via the sinks to alter the length of each segment to match with the segmentation clock output. See subsection “A reaction-diffusion model of axial patterning in the zebrafish”.

Phylogenetic character analysis of notochord induction/association of the chordacentrum– At several points the authors detail current thinking of chordacentrum involvement in patterning of the vertebra column and formation of the centra. It would be important, and clarifying, if the authors discuss classical work in basal teleosts such as Amia and Gar suggesting that notochord induction/association of a chordacentrum is ancestral in teleosts (Schultze and Arratia papers in the 1980s) and addressed in Laern, (1976). How much of the discordance between models/species that the authors mention represents different mechanisms or, alternatively, similar mechanisms studied at different levels of analysis?

This is an extremely interesting point, but we feel that an extensive discussion about this issue goes beyond the scope of this paper, simply because it would take a lot of space to cover this topic. Concerning the relevant remark of this reviewer, we have discussed some aspects of this issue with an expert in the field (PE Witten, Univ Ghent). We have included now a paragraph that cites the Arratia and Laerm papers and makes the statement that in teleosts chordacentrum formation is conserved, and that chordacentrum formation is an evolutionary novelty within Actionpterygians. As the reviewer suggests, many of the possible discrepancies and question marks concerning homology of chordacentrum/centrum formation could well be (and likely are) due to different levels of analysis and use of different methodologies. Only the availability of the *entpd5* transgenic line, an *entpd5* mutant and histology (AR stain) has allowed us to make strong statements about the situation in zebrafish. We find it difficult to make statements with the same rigor about other species.

Reviewer #2:In this study Forero et al., investigate the role of the segmentation clock in patterning chordacentra in zebrafish. Using a family of segmentation clock mutants, they disrupt segmentation to varying degrees and measure the chordacentra pattering. They propose a model in which the periodic patterning of chordacentra arises from a pattering process that can function independently from the segmentation clock. However, the somite patterns can modify the chordacentra patterning mechanism.My major issue is that the model ought to generate periodic patterning of chordacentra in the absence of somitogenesis clock input. If the proposed model can do this robustly then some elementary exploration of this case is necessary (i.e. a minimum requirement is to provide enough detail, so the results are reproducible and verify that there is a robust patterning mechanism in the case of no sinks). This validation is crucial as later the authors use the model to explain the location of activator peaks relative to sinks.

We thank the reviewer for the careful evaluation of our results, comments and suggestions. We completely agree with the reviewer that a key requirement to the theory is that there should be an autonomous patterning mechanism which in the absence of sinks generates a sequential segmented pattern from anterior to posterior. As we argue below this is indeed the case for the theory we propose here. We hope that, with the changes and additions to the manuscript described in detail in our reply, this point is clear and leaves no doubt.

1) The authors claim that 'in the absence of any sinks, the notochord would in our model segment without any defects with a periodicity determined entirely by the internal dynamics.'The key figure supporting this statement (Figure 14E) indicates that the mathematical model produces spatially periodic patterns at steady state in the absence of dermatome signal (i.e. s(x)=0 for all x).

This is indeed the case. In the absence of sinks, there is an autonomous reaction diffusion system that generates a sequential pattern of segments, starting from a small random perturbation localized at the anterior. The wavelength of the resulting pattern is regular and defined by the autonomous dynamics of the reaction diffusion system. We previously showed this result as Figure 15E. To acknowledge the importance of this aspect of the theory we now reproduce this as panel A of Figure 7, which is part of the main text in the revised manuscript. We also emphasize this aspect in the text before introducing the idea of sinks and the description of the wild type condition.

It is not adequately explained in the text how the proposed model does this. What is the patterning mechanism?A standard way to analyse the model is to consider behavior without diffusion. What are the steady states and what is their linear stability?My analysis suggests that in the absence of diffusion and with s_0_<1-d (i.e. small influence from the dermatome) there is a unique steady state (0,0) that is stable.When s_0_>1-d the origin becomes unstable and there are two non-zero steady states (i.e. presumably the model becomes bistable in this regime).

This is correct. The spatially homogeneous theory displays a bifurcation at s_0_ = 1-d, where a single stable fixed point gives rise to two stable fixed points separated by an unstable fixed point. In the absence of sinks we set s_0_ = 0 and find that d = 1 marks the onset of bistability. For d < 1 the system has a single stable fixed point. Motivated by the reviewer questions we decided to include this calculation, together with plots of the nullclines, in the Theory section in Materials and methods section. For conditions in which there is a single stable fixed point, like the ones considered in the manuscript, autonomous patterning is caused by diffusion driven destabilization of an otherwise stable homogeneous state.

This analysis is consistent with the authors numerical results; the spatially homogeneous steady state is destabilized in simulations where the sink strength is greater than 0.5. Outside of this region the spatially homogeneous steady state is monostable.Given the case where s_0_=0 is stable, the question then is whether the introduction of diffusion could cause an instability (e.g. Turing) and hence periodic patterning. If this is the case the authors should show it. However, even if this were true the wavefront behavior presented by the authors is nontrivial.

Again this is correct: it is differential diffusion that destabilizes the initially homogeneous state and leads to the formation of the periodic pattern in the absence of sinks with s_0_ = 0. The nullcline plots show that the type of instability that we can expect, as we now explain in more detail in the Theory section. We agree with the reviewer that the wavefront behavior is nontrivial.

I have tried to reproduce Figure 14E with my own code and as many of the details provided but cannot.

We hope that with the more detailed description of the methods and parameters we provide our results are now easier to reproduce.

Furthermore, in the interest of transparency we provide with the revised version of the manuscript the Python code that we use to generate the data, movies and snapshots.

We hope that the comments in the code make it clear. We run it within the Spyder environment, which comes with the Anaconda Python distribution, all free and open.

2) The PDE model has a parameter that is discontinuous in space. The authors ought to provide details of their discretisation scheme so that the reader can assess how they have dealt with this discontinuity.I suggest the following improvement: define the sinks independently of the numerical mesh and then approximate the steep switches with a continuous function such as tanh. In this way the sink strength parameter can be guaranteed to be continuous.

Concerning the steep sink switches, we had also tried continuous sink profiles like combinations of trigonometric functions and combinations of tanh functions. Since these smooth sink profiles produced qualitatively similar results to the steep sink profiles we settled for the latter in our first submission since they allowed for faster computation when it comes to statistics.

In the revised manuscript, we have chosen to follow the reviewer suggestion and recalculated everything using a smooth sink profile built from a combination of tanh functions, as described in the Theory section. The parameters of single sink shape are its width and steepness. The details of implementation can also be checked in the code provided.

In the numerical results reported in the revised manuscript, we use sinks that are wider that in the original version of the manuscript. We chose wider sinks to allow for smoother changes in the sink profile. For this reason, in the description of the different conditions reported in Figure 7, other parameters like sink strength and wavelength noise also change in the revised manuscript.

We provide full details of the discretization scheme in the Theory section of the Materials and methods section.

3) Stability analysis – is the pattering mechanism robust to small perturbations? It is worrying that the authors initialise on an unstable steady state of the homogeneous problem. Hence an infinitesimally small perturbation from these initial conditions could result in completely different model behaviour.Could the authors add some small amplitude noise to the initial conditions and present some numerical results. Is the proposed wavefront solution stable?

Concerning initial conditions, we describe in the Theory section how we implement a smooth concentration step localized to the anterior, with a randomly chosen amplitude for both components.

With the parameter choice that we make, the homogeneous system is stable for vanishing concentrations of the activator and the inhibitor. That is, if we start with no activator and no inhibitor, we stay that way in a homogeneous system. How does the pattern begin to form?

One possibility is to start with small random perturbations distributed all along the notochord. In the presence of diffusion, the homogeneous state loses stability and a pattern may form out of this initial random background fluctuation. We have tried this classical scenario, and what happens is that the whole system segments all at once, with all segments forming simultaneously. Segment formation is robust in this scenario and can accommodate a broad range of sink profile wavelengths. However, this is at odds with the experimental observation that ENTPD5 segments form sequentially.

One key feature of the experimental data is that ENTPD5 segments form sequentially, from anterior to posterior. It is interesting that it does this even in the presence of cues by the segmentation clock output: somitogenesis has finished essentially after the first day post fertilization, while notochord ENTPD5 patterning takes about 20 days to complete.

Both the autonomous pattern mechanism, here proposed as a reaction diffusion system, and the cues provided by the segmentation clock have to be consistent with this key observation.

One way to obtain a sequential segment formation is to start with an initial perturbation localized at the anterior and vanishing concentrations all across the rest of the notochord. From a biological perspective, since the anterior of the vertebrate axis is always more developmentally advanced than the posterior, such an anterior perturbation is a plausible hypothesis.

For this to work it is important that the reaction diffusion system does not have any local source terms, since this would trigger patterning all across the notochord at the same time. For this reason, we speculated that the cue from the segmentation clock might come in the form of a sink term, for example as local degradation of one or both the components, instead of a source term. This ensures that patterning is not triggered until the wave of activator and inhibitor arrives at a given point.

We are aware, and we agree with the reviewer that in this scenario, small perturbations might trigger segment formation before the wavefront arrives. Actually, this might be the case of the fill-in responses we observe in the data for some experimental perturbations, although we do not speculate in this respect.

4) The use of Fitzhugh Nagumo ought to be justified. I am not suggesting that the model needs to be linked to a molecular detail but some insight into the various terms would be helpful. The authors should describe the model assumptions and how they might be relevant to this system.

We thank the reviewer for this suggestion. We now describe the different terms in the reaction diffusion system in the Theory section. We argue that this particular model is a good choice for its simplicity and generality, with a single nonlinear term that allows for nontrivial fixed points to be stable. We have chosen to start with a full version of the theory and show how we non-dimensionalize the equations to arrive at the dimensionless form that we use in the manuscript. This gives the reader a good grasp of what parameters are in the different terms and what the assumptions are in this dimensionless theory. We also perform the linear stability analysis of the homogeneous system as described above. We hope that this better justifies the use of this model and highlights its features.

5) “In the simulations of fss, guu, and fum this strict correspondence between sink and activator is lost (Figure 7B and C); activator peaks occur both together with sinks and in between them.”Can the authors use the model to provide insight into how this can happen? Is this observation a generic feature of activator-inhibitors models? If it is generic, then showing results from other reaction-diffusion models would help. If it is not generic, then the properties of the proposed model that yield the interesting behaviour ought to be defined and investigated more thoroughly.

There is a range of wavelengths that an activator-inhibitor model can support when diffusion drives it out of the homogeneous state. If the sink ahead of the segmentation wave is too close or too far away in a way that compromises the supported wavelengths, then a skipping or intercalation may occur. The existence of a range of wavelengths which are supported is true for all activator-inhibitor models, so we expect sink skipping or segment intercalation to be generic features. The FitzHughNagumo model that we explore is very generic and representative of a large class of systems that can be mapped to this kind of nonlinear dynamics near the bifurcation of a single steady state into two stable states.

6) Given the mathematical model takes up almost two pages of the results then I suggest that a figure exploring the relevant features of the model is appropriate.

We have followed the reviewer suggestion, which is in agreement with reviewer #1. We decided to include Figure 7 in the main text of the revised version of the manuscript. In this figure, we added a top panel showing the behavior of the theory in the absence of sinks, given the key importance of this aspect of the theory as pointed by the reviewer in (1).

We thought that a more thorough exploration of the theory diverts from the main focus of this work, and therefore opted to keep Figure 7—figure supplement 2 showing the behavior of the model as the sink profile is changed in different ways as a supplementary figure. Figure 7—figure supplement 4 compares statistics from experiments and simulations, and new Figure 7—figure supplement 5 illustrates nullclines and stability.

[Editors' note: further revisions were requested prior to acceptance, as described below.]

Reviewer #1:In the text the authors often list fss, her1;her7, and tbx6;her1;her7 mutant combinations. I assume the fss allele of tbx6 is the one being used (unless another has been generated). If this is the case the text should reflect this as tbx6-/-. her1; her7, tbx6;her1;her7 mutants. The authors have correctly labeled this in the figures, but not the figure legends nor text.

Thanks for pointing this out - we have made all these changes.

Somewhere in the text the authors should address whether these alleles are thought to be null or strong loss-of-function. Data is not needed, rather citation of previous genetic analysis on available alleles and a statement in the text is just helpful.

The alleles used in this study are either known to be effectively null or expected to be strong loss-of-function. We have added several sentences to describe what is known about the mutations in the first paragraph of the Results section.

Results section “A reaction-diffusion model of axial patterning in the zebrafish”: “intrinsic segmentation mechanism, likely sheath cells" Sheath cells is not a mechanism. Do the authors mean within sheath cells?

Thank you, we now state “within” the sheath cells.

Reviewer #2:1) Whilst the theory section has been improved, now that it is explicit that the authors are proposing the Turing mechanism as the underlying patterning mechanism, can they provide a fuller analysis of the unstable wavenumbers for the parameter values presented in the simulations? It is important to characterize how the unstable wavenumbers (and corresponding wavelengths) relate to the typical inter-sink distance (e.g. presumably the model parameters have been chosen to give a wavenumber that is approximately of the same order as the inter-sink distance).

Characterization of the unstable wavenumbers is a good general technical question one could ask about any similar model. The answer to the particular question asked by the reviewer is already there in Figure 7A; in the sink-less simulations, the natural wavelength for the invading wave is smaller than that of the wild type sink wavelengths. Characterizing this relationship more fully, as requested, is technically possible, but we argue that since this would only hold for one particular set of parameters, and not provide a general picture, it is not clear to us that it brings a strong addition. Furthermore, the outcome of this exercise would not affect the conclusions of the paper. We now point out explicitly that the natural wavelength for the invading wave is close to that of the wild type sink wavelength in the text in subsection “A reaction-diffusion model of axial patterning in the zebrafish”.

2) As the authors did not present the results with arbitrarily small noise in the initial data as I requested, I have solved the equations myself and attach the results.

Thank you for the comment – we had also done this previously. However, due to the direction and scope of what we aim to do here, we argue that it is not appropriate for inclusion in this paper.

In Noise.mp4 I solve the model as it is presented in the paper. Note that the key qualitative behaviour is that independent of somite signal, a propagating wavefront leaves a periodic pattern in its wake. This is the behavior observed experimentally that any reasonable model ought to replicate.

We believe that the reviewer means NoNoise.mp4, which is indeed the behaviour that our model replicates as shown in Figure 7 and Figure 7—figure supplement 1 and Figure 7—figure supplement 6A.

In NoNoise.mp4 I have added very low amplitude noise to the initial conditions. Note that the noise destabilises the wavefront solution and the domain patterns simultaneously rather than sequentially.These numerical results indicate that even the addition of infinitesimally small noise throughout the domain results in the wavefront solution being lost. Given the presence of noise in biological systems, this robustness issue is a fundamental limitation that, by not addressing in the main text of the manuscript, the authors seem to have neglected.

This comment concerns the robustness of the model to a particular kind of noise. The reviewer rightly points out that adding noise to the initial spatial domain breaks the wave-like invasion of the mechanism. We are of course aware of the presence of noise in biological systems, and also the effect of small noise amplitude on the solution. We now include reference to the issue of robustness the Main Text in subsection “A reaction-diffusion model of axial patterning in the zebrafish” and point the reader to the revised Materials and methods section for a fuller discussion.

In this paper, given that we still lack the molecular mechanism or details, our overall goal is to present as simple a picture for the segmental gene expression phenomenon as possible. This necessitates an abstraction of the problem and an idealization of the biological context. There are several ways to extend or adapt the model to make its wavefront solution stable to noise, yet these introduce additional complexity (see below), which we have no biological data support for. Also, we are concerned that making the model more complex may distract from the main conclusions of the paper, while not altering them.

I suggest the following:

i) The authors build a convincing case, with reference to the pattern formation literature, that deals with the robustness issue. i.e. are there other published examples of wavefront propagation mediated spatial patterning with a similar lack or robustness?

We are not aware of any in the literature. Nevertheless, we note that the absence of a previous example in the literature is not a strong test for the validity of a new proposal.

orii) The authors build on their proposal that robustness could be mediated by a maturation gradient. This could be incorporated into the model by considering a competency domain where the authors solve the reaction diffusion equations on some domain [0, s(t)] where s(t) is a suitably chosen function of time.Such a model would have a fundamentally different behaviour in that the imposed wavefront would determine the speed of segmentation rather than the Turing instability. Moreover, it would be robust to infinitesimally small noise.For the suggested approach the authors could see, for example, Madzvamuse et al., 2005 or Crampin et al., 2002.

This is an interesting suggestion, but the case in the embryo is probably not well-represented by a growing domain, as suggested by the reviewer. This is because the notochord has reached a length of some millimetres by the time the metameric entpd5 expression domains start to be formed in a wave-like manner along the axis. Although the axis does continue to elongate during this interval, the velocity of the entpd5 expression wavefront is many times higher. To first approximation, growth of the domain is almost certainly not driving the patterning. An alternative approach might be to model a maturation gradient using a time-dependent change in the parameters. However, we argue that although this modification would also cause the wavefront to become robust to noise, it would involve a number of choices about the form of the hypothetical wavefront, about which we have no experimental data. In any case, it is important to point out that a no-noise initial condition as we use, or some form of trigger of the wavefront as suggested by the reviewer, can be viewed as alternative effective descriptions that produce sequential pattern formation. Although these are different in terms of the model structure and properties, in the absence of any experimental data, adding in a new structure is not parsimonious. We argue that pursuing this line of investigation is of great interest, particularly when new experimental evidence comes to light about the molecular details but is well beyond the scope of the current manuscript.

We appreciate the constructive criticism of the model, and indeed these comments have addressed potentially misleading ambiguities in our presentation. We have now included a more extensive discussion of initial conditions in the Materials and methods section, and the caveats or limits to the current description as well as potential approaches that could offer a way forward. We also refer to this explicitly in the Main Text in subsection “A reaction-diffusion model of axial patterning in the zebrafish”.